# MOSAIC-IT: FREE COMPOSITIONAL DATA AUGMENTATION IMPROVES INSTRUCTION TUNING

## ABSTRACT

Finetuning large language models with a variety of instruction-response pairs has enhanced their capability to understand and follow instructions. Current instruction tuning primarily relies on teacher models or human intervention to generate and refine the instructions and responses for training, which are costly, non-sustainable, and may lack diversity. In this paper, we introduce Mosaic Instruction Tuning (Mosaic-IT), a human/model-free compositional data augmentation method that can efficiently create rich and diverse augmentations from existing instruction tuning data to enhance the LLMs. Mosaic-IT randomly concatenates multiple instruction data into one and trains the model to produce the corresponding responses with predefined higher-level meta-instructions to strengthen its multi-step instruction-following and format-following skills. Our extensive evaluations demonstrate a superior performance and training efficiency of Mosaic-IT, which achieves consistent performance improvements over various benchmarks and a $80\%$ reduction in training costs compared with original instruction tuning. Our codes and data are available at `https://anonymous.4open.science/r/mosaic-955B`.

## 1 INTRODUCTION

The emergence of Large Language Models (LLMs) Brown et al. (2020); Scao et al. (2022); OpenAI (2023); Touvron et al. (2023a;b); Jiang et al. (2023) along with their remarkable performance in down-stream tasks Zhao et al. (2023); Xu et al. (2024a), has revolutionized the domains of Artificial Intelligence and Natural Language Processing. A key component of the recipe to unlock the exceptional ability of LLMs in understanding and following instructions is the technique of Instruction Tuning (IT) Mishra et al. (2021); Wei et al. (2022); Chung et al. (2022); Wang et al. (2023c); Zhang et al. (2023); Xu et al. (2024a), which involves the fine-tuning of LLMs on datasets comprising corresponding instruction-response pairs.

To ensure the quality of instruction tuning data, earlier efforts Brown et al. (2020); OpenAI (2023); Touvron et al. (2023a); Jiang et al. (2023) carefully curate extensive, diverse, and high-quality datasets manually. Although these datasets encompass a wide range of instructions to improve instruction tuning, they require the responses to be meticulously curated by human experts Khashabi et al. (2020); Ye et al. (2021); Wei et al. (2022); Wang et al. (2022); Du et al. (2022). Alternatively, some approaches Wang et al. (2023b); Taori et al. (2023); Xu et al. (2023); Li et al. (2023a) leverage more capable teacher LLMs to reduce the labor-intensive process of data generation. For example, the Alpaca Taori et al. (2023) utilizes self-instruct Wang et al. (2023b) to automatically generate diverse instruction tuning datasets, and the WizardLM Xu et al. (2023) proposes to complicate the existing instruction data by an evolution algorithm. Building on this trend and the widely acknowledged notion that more complicated instructions are more beneficial for LLMs' instruction-following ability Xu et al. (2023); Zhao et al. (2024), numerous strategies Zhao et al. (2024); Wu et al. (2024); Ding et al. (2023); Li et al. (2023a); Liu et al. (2023a); Li et al. (2024b;a); Guo et al. (2024); Xu et al. (2024a) have been proposed to further diversify and complexify the instruction-response pairs, utilizing teacher models like ChatGPT-3.5 and GPT-4 OpenAI (2023).

Despite the enhanced performance in instruction-following ability offered by these existing methods, they face **Two** major issues: (1) They heavily rely on teacher models or human annotators to rewrite instruction-response pairs, which highlights the resource-intensive nature and their constraints on scalability; (2) They only increase the complexity within the scope of a single

instruction, which limits the potential improvement in LLMs' instruction-following capabilities. Motivated by the Dense and Aligned Captions Doveh et al. (2023) proposed for vision language (VL) models and the mosaic data augmentation proposed in Yolov4 Bochkovskiy et al. (2020), we hypothesize that denser instructions benefit the LLM alignment, i.e. the process of instruction tuning should not be constrained by one single instruction but be extended to follow several instructions at a time, which represents a higher level of instruction-following ability that is beneficial to the training process. A similar concept during the inference phase is proposed by batch prompting Cheng et al. (2023); Lin et al. (2024), where multiple samples are grouped in one batch allowing LLMs to generate multiple responses at one inference, while its performances are sub-optimal. Moreover, our preliminary experiments on *GPT-3.5-turbo* and *GPT-4-turbo* show that even for these strong proprietary LLMs, their performances degrade dramatically if required to follow several instructions at one time, the experimental results are presented in the Section 5 Further Discussion. Thus, these performance degradation phenomenons indicate the complexity of this setting and the necessity of further training for this higher-level capability.

As orthogonal to the existing instruction tuning methods, we introduce Mosaic Instruction Tuning (Mosaic-IT), an innovative and model/human-free compositional approach that augments existing instruction tuning datasets, which concurrently improves the LLM performances and lowers the training expenses. As shown in Figure 1, in our method, multiple instructions and corresponding responses from the original dataset are concatenated into a single sample for fine-tuning, simulating the multi-instruction-following scenarios at no cost. Without applying any additional strategies, we term this simple process as the **Primary Mosaic Strategy**. We posit that this mosaic strategy process significantly improves the complexity and density of the original instructions, learning from which directly benefits LLMs in their instruction-following ability. Additionally, this method offers the advantage of directly reducing the total count of instruction-response pairs, thereby cutting down on training iterations,

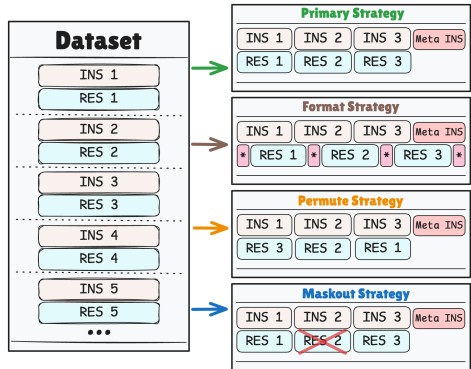

Figure 1: The illustration of our Mosaic-IT with different strategies. Given the original dataset, our method randomly samples and concatenates them together into more complex samples, simulating the multi-instruction-following scenarios at no cost.

and accelerating the training process significantly by approximately 80% reduction.

Though effective, the Primary Mosaic strategy constrains LLMs in responding to the instructions in the original order and format, potentially limiting its further potential. Thus we further introduce three **Advanced Mosaic Strategies** aimed at enhancing the diversity and complexity of the mosaicked instruction-response pairs: **Format**, **Permute**, and **Maskout**, in which an additional meta-instruction is provided as a higher-level guideline for LLMs to follow the given instructions. Illustrative examples are presented in Figure 2. Specifically, in the Format strategy, some arbitrary parsing formats will be defined in the meta-instruction thus forcing LLMs to follow these formats, which notably enhances the LLMs' capacity to follow formats. In the Permutation strategy, an arbitrary permuted order is defined thus forcing LLMs to respond in a desired order. In the Maskout strategy, some arbitrary instructions are sampled which meta-instruction forces LLMs to ignore. Moreover, the use of these Advanced strategies not only boosts the performance in several evaluation metrics but also keeps our method free of additional costs.

In summary, our primary contributions can be illustrated as follows:

- We propose a novel human/model-free data augmentation method, **Mosaic-IT**, which extends existing instruction tuning from handling one single instruction at a time to following multiple instructions in diverse forms. This approach significantly enhances the potential utilization of existing high-quality datasets.

- Mosaic-IT improves the instruction-following abilities of LLMs compared to training on original data, as evidenced by consistent performance gains across a wide range of benchmarks, model families, and datasets, demonstrating strong generalization capabilities.

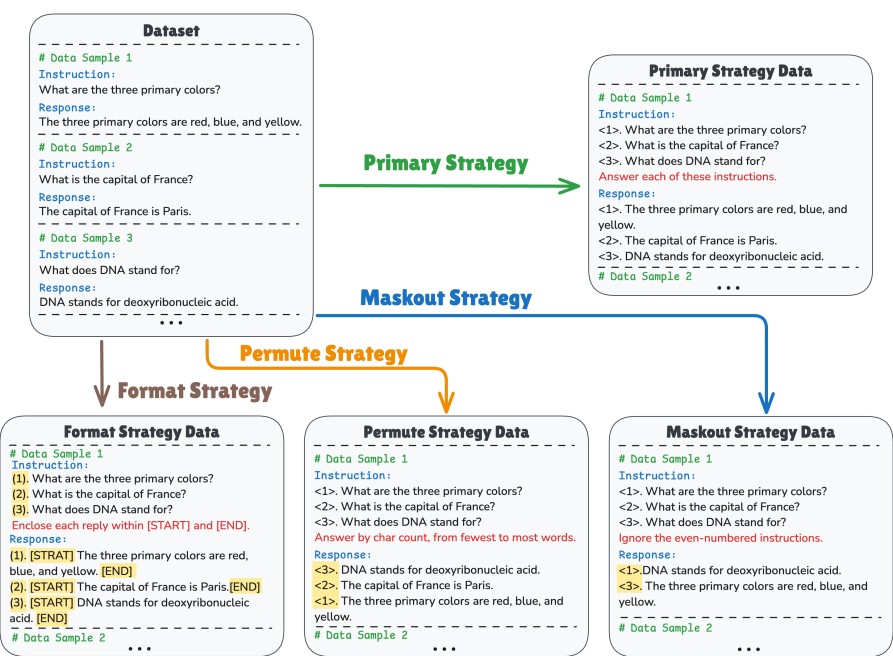

Figure 2: Illustrative examples of Mosaic-IT. Given 3 simple data points, our method can concatenate them into overall data samples with diverse forms. Texts in red represent the meta-instructions that define the formats or orders for LLMs to respond. Texts in yellow are major response differences of each strategy. The **Primary Strategy** only concatenates data together. The **Format Strategy** requires LLMs to respond in predefined formats. The **Permute Strategy** requires LLMs to respond in specific orders and the **Maskout Strategy** requires LLMs to ignore some of the instructions.

- Mosaic-IT substantially increases training efficiency by reducing the required number of training iterations, resulting in an approximate 80% reduction in training time, as confirmed by experimental results.

## 2 METHODOLOGY

### 2.1 PRELIMINARIES

The instruction tuning dataset, defined as $D$, consists of $n$ data samples, each represented by a triplet $(Instruction, Input, Response)$. For simplicity, we define $x = \text{map}(Instruction, Input)$ as the unified instruction, and $y$ as the corresponding response. Therefore, $D$ can be represented as $(x_1, y_1), (x_2, y_2), \ldots, (x_n, y_n)$, denoting a set of $n$ instruction-response pairs. Let $p_\theta(\cdot)$ denote the LLMs to be trained, with parameters $\theta$. In the instruction tuning setting, $p_\theta$ is typically fine-tuned by maximizing the following objective on each data $(x_i, y_i)$, $y_{i,j}$ represents the $j_{th}$ token of response $y_i$, $y_{i,<j}$ represents the tokens prior to $y_{i,j}$, and $l_i$ represents the token length of $y_{i,j}$:

$$\max_\theta \sum_{j=1}^{l_i} \log p_\theta \left( y_{i,j} | x_i, y_{i,<j} \right), \tag{1}$$

### 2.2 MOSAIC-IT

Motivated by the success of the existing data-centric instruction tuning methods, a line of approaches is proposed to further enhance the instruction-response pairs utilizing extra teacher LLMs Xu et al. (2024a). Though effective, all existing methods for instruction tuning restrict training samples to just one instruction, which severely limits the potential of the existing high-quality data and the instruction-following ability of the models to be trained. Motivated by the Dense and Aligned Captions Doveh et al. (2023) for VL, we hypothesize that denser instructions benefit the LLM alignment, thus the process of instruction tuning should not be constrained by one single instruction but be extended to follow several instructions at a time, which represents a higher level of instruction-following ability that is beneficial to the training process. Thus, we propose Mosaic Instruction Tuning (Mosaic-IT) as shown in Figure 1.

### 2.2.1 PRIMARY MOSAIC STRATEGY

Exploring the concept of concatenating random instruction-response pairs into a unified instruction-response pair for training remains largely unexplored. The primary challenge lies in crafting a coherent overall instruction and obtaining its corresponding response. Most existing methods utilize a strong teacher model to rewrite and polish the instructions with prompting techniques and generate corresponding responses, introducing more cost by actually re-generating new data samples. To harness the full potential of existing data rather than directly discarding them, we introduce a simple compositional approach as shown in Figure 2, in which instructions are randomly concatenated with serial digits to form an *overall instruction*. The concatenated overall instruction is denoted as $[x_1, ..., x_k]$, with the corresponding overall response concatenated as $[y_1, ..., y_k]$. Here, $k$ denotes the number of original data samples integrated into each overall sample.

In this framework, the fundamental instruction-following capability is triggered by the existing instruction-response pairs, and the mosaic strategy extends this capability to a higher level in which LLMs are forced to follow multiple instructions. It represents a much more complicated scenario that benefits LLMs compared with traditional single-task instructions. Consequently, the objective function for each concatenated overall data sample can be formulated as follows:

$$\max_{\theta} \sum_{j=1}^{l} \log p_{\theta} \left( [y_1, ..., y_k]_j | [x_1, ..., x_k], [y_1, ..., y_k]_{<j} \right), \qquad (2)$$

Here, $[y_1, ..., y_k]_j$ denotes the $jth$ token of the overall response, $[y_1, ..., y_k]_{<j}$ denotes the tokens prior to $jth$ token, and $l$ represents the length of overall response. This formulation encapsulates the essence of our approach, optimizing the model parameters $\theta$ to maximize the likelihood of generating the correct sequence of responses for the given overall instruction.

### 2.2.2 ADVANCED MOSAIC STRATEGIES

Though effective, this simple primary mosaic strategy constrains LLMs in responding to the instructions with the original order and format, potentially limiting its generalization and practical usage. In our method, the instructions and corresponding responses from the original dataset can be viewed as atomic components and our method randomly combines these elements together to form new instructions and responses. This nature allows us to further complicate this process with fancier strategies thus forcing LLMs to follow more complicated overall instructions. Hence, we propose three **Advanced Mosaic Strategies** to complicate and diversify the mosaicked samples as shown in Figure 2, including Format, Permute, and Maskout, with meta-instructions guiding them.

**Format** In the Format strategy, some arbitrary formats are defined in the meta-instruction to force LLMs to follow these formats in the response. The formats mainly contain two categories: 1) *Serial Digit Format* and (2) *Response Parsing Format*. The serial digits establish the initial instruction order that guides LLMs to follow sequentially. We manually define 10 types of serial digit format, which will be randomly sampled during each mosaic process. For response parsing, we simulate the scenario where the users try to extract specific information from the responses. We define 27 types of parsing brackets and 17 types of parsing text pairs, which will be randomly sampled and assembled during each mosaic process. Examples can be found in Appendix D, which can be easily extended for customized training settings. We denote responses with specific formats as $y_i' = wrap(y_i, s_{format})$, and $l$ as the token length of the overall response. An additional meta-instruction $s_{format}$ specifying the required format will be included in the overall instruction. Thus, the objective function for each mosaic data point:

$$\max_{\theta} \sum_{j=1}^{l} \log p_{\theta} \left( [y_1', ..., y_k']_j | [x_1, ..., x_k, s_{format}], [y_1', ..., y_k']_{<j} \right) \qquad (3)$$

**Permute and Maskout** Building upon the Format strategy, we further introduce two strategies for our Mosaic-IT, Permutation and Maskout.

In the **Permute** strategy, an arbitrary permuted order is defined in the meta-instructions, forcing LLMs to follow. Moreover, several high-level rules are defined to ensure the complexity and diversity of meta-instructions, e.g., forcing LLMs to respond to each instruction in the randomly

generated permutation list, forcing LLMs to respond in the alphabetical order of each instruction, forcing LLMs to respond according to the length of instructions, etc. The detailed rule types and descriptions are depicted in Appendix D. These various meta-instructions not only provide higher-level guidelines for LLMs to follow multiple instructions but also inherently enhance the instruction perception ability of LLMs. In our settings, LLMs are required to generate responses selectively conditioned on some critical parts of the overall instruction, forcing them to first understand the formats and other requirements, indicating a more comprehensive understanding of the context given. The meta-instruction is denoted as $s_{permute}$ and is included in the overall instruction. The permuted response list is denoted as $[y'_{1'}, ..., y'_{k'}] = Permute([y'_1, ..., y'_k], s_{permute})$. Thus the objective function can be formulated as below:

$$\max_\theta \sum_{j=1}^{l} \log p_\theta \left([y'_{1'}, ..., y'_{k'}]_j | [x_1, ..., x_k, s_{format}, s_{permute}, [y'_{1'}, ..., y'_{k'}]_{<j}]\right), \tag{4}$$

In the **Maskout** strategy, some arbitrary instructions are selected in the meta-instructions forcing LLMs to ignore them. Several high-level rules are also defined similarly to the permute strategy, including forcing LLMs to ignore the instructions with given random digits, forcing LLMs to ignore the longest one/several instructions, forcing LLMs to ignore odd-numbered instructions, etc. The details are provided in Appendix D. Similarly, the meta-instruction is denoted as $s_{maskout}$ and the response list is denoted as $[y'_1, ..., y'_m] = Maskout([y'_1, ..., y'_k], s_{maskout})$, where $m$ is the count of responses after masking out. Thus the objective function can be formulated as below:

$$\max_\theta \sum_{j=1}^{l} \log p_\theta \left([y'_1, ..., y'_m]_j | [x_1, ..., x_k, s_{format}, s_{maskout}], [y'_1, ..., y'_m]_{<j}\right) \tag{5}$$

It's important to note that our mosaic strategies entail **no supervision cost**, and the predefined rules are flexible and have the potential for further extension. We utilize the version with three Advanced strategies as our default Mosaic-IT.

**How to decide the Number of Instructions** $k$: Number of Instructions denotes the number of original data samples that are integrated into an overall sample. In addition to the detailed mosaic strategies being used, this count also dramatically affects the effect of Mosaic-IT. Our experiments reveal that larger and more diverse numbers of instructions will benefit LLM training. By default, we set the maximum number of instructions as $k_{max} = 10$, and randomly sample an integer that is smaller or equal to $k_{max}$ under a uniform distribution. If the number causes the data sample to be longer than the max length, it will be automatically reduced to the max number which remains the sample length within the limits.

## 3 EXPERIMENTAL SETUP

### 3.1 IMPLEMENTATION DETAILS

The experiments are conducted on: Llama2-7B, Llama2-13B Touvron et al. (2023b), and Mistral-7B Jiang et al. (2023), Llama-3-8B Dubey et al. (2024), Phi-3 Abdin et al. (2024), and Gemma2-2B Team et al. (2024). The training datasets include Alpaca Taori et al. (2023), Alpaca-GPT4 Peng et al. (2023), WizardLM Xu et al. (2023), Vicuna 1M Zheng et al. (2024a), and Magpie Xu et al. (2024b) datasets. Due to the really large size of Vicuna 1M and Magpie, 300k instances are randomly sampled for our experiments. The detailed description of datasets and the training configurations are introduced in Appendix B.

### 3.2 EVALUATION METRICS

We utilize **five** automatic evaluation metrics, including (i) LLM-based Pair-wise Comparison, (ii) Open LLM leaderboard, (iii) MT-Bench, (iv) Alpaca Eval, and (v) IF Eval, and (vi) Human evaluation to verify the effectiveness of our method. They are widely accepted evaluation metrics for measuring LLMs' instruction-following capabilities. The introductions of the five automatic evaluation metrics are provided in Appendix B.

Table 1: The performance comparison on the Pair-wise Comparison Winning Score and the Open LLM Leaderboard, on 3 different base models and 3 different instruction tuning datasets.

| Model | Dataset | Method | Pair-wise ↑ Winning Score | Huggingface Open LLM Leaderboard ↑ | | | | |
|---|---|---|---|---|---|---|---|---|
| | | | | Average | ARC | HellaSwag | MMLU | TruthfulQA |
| Mistral-7B | Alpaca-GPT4 | Baseline | 1.000 | 59.70 | 55.03 | 78.87 | 56.01 | 48.88 |
| | | Mosaic-IT | **1.349** | **63.65** | 59.04 | 81.85 | 60.09 | 53.62 |
| | Alpaca | Baseline | 1.000 | 55.15 | 51.96 | 74.61 | 52.85 | 41.20 |
| | | Mosaic-IT | **1.390** | **58.86** | 56.23 | 79.57 | 57.06 | 42.58 |
| | Wizard-70k | Baseline | 1.000 | 57.86 | 51.88 | 77.93 | 53.76 | 47.89 |
| | | Mosaic-IT | **1.161** | **61.11** | 57.85 | 82.13 | 57.42 | 47.08 |
| Llama2-7B | Alpaca-GPT4 | Baseline | 1.000 | 58.71 | 54.69 | 80.05 | 47.89 | 52.21 |
| | | Mosaic-IT | **1.073** | **58.84** | 54.18 | 80.54 | 47.92 | 52.70 |
| | Alpaca | Baseline | 1.000 | 55.25 | 54.35 | 78.65 | 47.02 | 40.98 |
| | | Mosaic-IT | **1.096** | **55.32** | 53.75 | 78.65 | 46.88 | 41.98 |
| | Wizard-70k | Baseline | 1.000 | 57.09 | 54.18 | 79.25 | 46.93 | 48.02 |
| | | Mosaic-IT | **1.197** | **57.41** | 54.69 | 79.69 | 48.11 | 47.13 |
| Llama2-13B | Alpaca-GPT4 | Baseline | 1.000 | 61.47 | 58.70 | 83.12 | 54.13 | 49.92 |
| | | Mosaic-IT | **1.110** | **63.26** | 58.87 | 83.54 | 55.75 | 54.87 |
| | Alpaca | Baseline | 1.000 | 57.63 | 57.25 | 81.23 | 54.13 | 37.91 |
| | | Mosaic-IT | **1.046** | **58.80** | 56.57 | 81.79 | 54.28 | 52.55 |
| | Wizard-70k | Baseline | 1.000 | 61.24 | 57.04 | 83.39 | 55.76 | 48.78 |
| | | Mosaic-IT | **1.078** | **61.50** | 58.70 | 83.69 | 56.44 | 47.18 |

Table 2: The performance comparison on the MT-Bench, Alpaca Eval, and IF Eval Benchmarks. Rate(LC) in Alpaca Eval represents the length-controlled win rates. In IF Eval, Prompt, and Inst represent Prompt-level and Instruction-level accuracy; S and L represent Strict and Loose versions.

| Model | Dataset | Method | MT-Bench ↑ | | Alpaca Eval 2 ↑ | | IF Eval ↑ | | | |
|---|---|---|---|---|---|---|---|---|---|---|
| | | | 1-round | 2-round | Rate (LC) | Rate | Prompt (S) | Inst (S) | Prompt (L) | Inst (L) |
| Mistral 7B | Alpaca-GPT4 | Baseline | 6.44 | **5.26** | 3.98 | 7.28 | 32.53 | 42.93 | 35.86 | 45.92 |
| | | Mosaic-IT | **7.11** | 4.69 | **5.00** | **7.81** | **37.15** | **48.56** | **38.08** | **50.23** |
| | Wizard-70k | Baseline | 6.21 | **4.70** | 4.13 | 6.46 | 39.56 | 49.88 | 41.96 | 53.00 |
| | | Mosaic-IT | **6.95** | 4.32 | **4.44** | **7.56** | **40.85** | **51.80** | **45.47** | **56.47** |

**Human Evaluation** is further implemented to substantiate the superiority of our approach based on the WizardLM test set. The test set contains 100 samples randomly sampled from the original WizardLM test set. Three human evaluators were tasked with comparing the outputs generated by the models under consideration, using the same criteria as in the previous pairwise evaluation. Each evaluator was presented with three response options: Win, Tie, and Loss. The final outcomes were determined by a majority vote.

# 4 EXPERIMENTAL RESULTS

## 4.1 MAIN RESULTS

In this section, we present the evaluation results comparing our methods with the baseline methods on several baseline models (Mistral-7B Jiang et al. (2023), Llama2-7B Touvron et al. (2023b), Llama2-13B) and instruction tuning datasets (Alpaca-GPT4 Peng et al. (2023), Alpaca Taori et al. (2023), WizardLM-70k Xu et al. (2023)), on **Two** general evaluation settings (Pair-Wise Comparison and Open LLM leaderboard) described above, as shown in the Table 1. **Pair-wise Winning Score** indicates the result directly comparing our models with the corresponding baseline models, which is calculated as $(\text{Num(Win)}-\text{Num(Lose)})/\text{Num(All)} +1$. These values that are greater than 1.0 represent better responses generated by our models. The performances on the **Huggingface Open LLM Leaderboard** are also presented, and we bold the greater average values for each comparison. The consistent outperforming results on different base models and datasets represent the effectiveness and robustness of our methods.

To better understand how our method improves the instruction-following abilities of LLMs, we further compare the performance on other **Three** benchmarks for fine-grained analysis based on the Mistral-7B base model with two datasets as shown in Table 2. On the **MT-Bench**, the 1-round scores of our method are higher, indicating that our method mainly improves the response quality

Table 3: The performance comparison on more model families and datasets on all five automatic evaluation metrics. In IF Eval, P and I represent Prompt-level and Instruction-level accuracy.

| Model | Dataset | Method | Pair-wise ↑ Score | Open LLM ↑ Average | Alpaca Eval 2 ↑ Rate (LC) | Rate | MT-Bench ↑ 1-round | 2-round | IF Eval ↑ P(L) | I(L) |
|---|---|---|---|---|---|---|---|---|---|---|
| Llama-3-8B | Vicuna | Baseline | 1.000 | 52.51 | 2.15 | 1.36 | 6.70 | 5.06 | 21.26 | 33.45 |
| | | Mosaic-IT | **1.234** | **55.62** | **3.09** | **2.05** | **6.85** | **5.40** | **31.42** | **45.56** |
| | Magpie | Baseline | 1.000 | 56.15 | 9.22 | 13.74 | 8.10 | 7.08 | 35.67 | 47.72 |
| | | Mosaic-IT | **1.133** | **60.13** | **12.23** | **16.05** | **8.36** | **7.49** | **40.67** | **52.76** |
| Phi-3 | Vicuna | Baseline | 1.000 | 62.06 | 4.20 | 2.74 | 5.34 | 4.18 | 30.50 | **43.17** |
| | | Mosaic-IT | **1.083** | **62.30** | **5.95** | **3.83** | **5.89** | **4.53** | **32.35** | 41.85 |
| | Magpie | Baseline | 1.000 | 62.90 | 13.82 | **17.68** | 7.78 | **6.42** | 44.36 | 55.52 |
| | | Mosaic-IT | **1.014** | **63.54** | **14.04** | 17.67 | **7.89** | 6.16 | **50.83** | **62.35** |
| Gemma2-2B | Vicuna | Baseline | 1.000 | 48.90 | 1.72 | 1.31 | 6.69 | 5.25 | 23.66 | 35.61 |
| | | Mosaic-IT | **1.266** | **51.31** | **1.90** | **1.38** | **6.93** | **5.26** | **24.03** | **36.93** |
| | Magpie | Baseline | 1.000 | 46.37 | 5.35 | 7.77 | 4.57 | 3.23 | 21.81 | 32.49 |
| | | Mosaic-IT | **1.032** | **48.36** | **5.66** | **8.54** | **5.16** | **3.96** | **22.18** | **34.77** |

for single-round conversations, which is reasonable as the meta instructions only focus on single-round formats. On the **Alpaca Eval** benchmark, our method has a consistent improvement with or without the Length Control (LC), indicating that the improvement of response qualities does not directly originate from the length of responses. On the **IF Eval** benchmark, our method consistently improves the performances on all 4 different settings, both Prompt-level and Instruction-level, both Strict version and Loose version. Compared with the previous benchmarks, IF Eval mainly focuses on the constraint-following ability of LLMs. The consistent improvement in this benchmark represents that our method not only improves the response qualities of the LLMs but also improves their controllability regarding formats. Given that our method is a cost-free augmentation technique that does not rely on any additional models, the observed improvements are remarkable.

Moreover, to further verify the effectiveness of our method, more experiments on different model families and data families are conducted, as shown in Table 3, including Llama-3-8B Dubey et al. (2024), Phi-3 Abdin et al. (2024), and Gemma2-2B Team et al. (2024) models on Vicuna 1M Zheng et al. (2024a), and Magpie Xu et al. (2024b) datasets. For these two datasets, 300k data are randomly sampled to verify the scalability of our method when dealing with large amounts of instruction-tuning data. The performances of our models consistently outperform the baseline models across different model families and data sources, ranging from diverse data qualities.

Further **Human Evaluations** are conducted on Mistral-7B with Alpaca-GPT4 and WizardLM dataset. For the comparison on (1) Alpaca-GPT4: the model using Mosaic-IT wins on 68 out of 100 instruction, ties on 3, and losses on 29 instructions; on (2) WizardLM: the model using Mosaic-IT wins on 63 out of 100 instruction, ties on 6, and losses on 31 instructions. This human evaluation also further verifies the effectiveness of our Mosaic-IT.

## 4.2 Ablation studies

In this section, extensive ablation experiments are conducted on Mistral-7B using with the Alpaca-GPT4 dataset to verify our method. We utilize Pair-wise comparison for evaluation.

Table 4: Ablation on (a) Mosaic-IT strategies and (b) Max Number of Instructions.

(a) Ablation on Mosaic-IT strategies.

| | Winning Score | Win | Tie | Lose |
|---|---|---|---|---|
| Primary | 1.261 | 110 | 55 | 53 |
| Format | 1.284 | 109 | 62 | 47 |
| Permute | 1.334 | 118 | 55 | 45 |
| Maskout | 1.376 | 121 | 58 | 39 |
| Permute/Maskout | 1.349 | 123 | 48 | 47 |

(b) Ablation on the Max Number of Instructions.

| | Winning Score | Win | Tie | Lose |
|---|---|---|---|---|
| Max Count = 2 | 0.989 | 70 | 75 | 73 |
| Max Count = 4 | 1.142 | 92 | 65 | 61 |
| Max Count = 6 | 1.303 | 111 | 62 | 45 |
| Max Count = 8 | 1.294 | 112 | 58 | 48 |
| Max Count = 10 | 1.349 | 123 | 48 | 47 |
| Max Count = 12 | 1.376 | 124 | 52 | 42 |

**Ablation on Mosaic Strategies** is presented in Table 4a. *"Primary"* represents the Primary Mosaic Strategy. The winning score of this setting is greater than 1.0, indicating a better performance compared with the baseline method. This comparison directly verifies the effectiveness of the idea of introducing multiple instructions during training, which complicates the instructions at no cost and improves the instruction-following ability of LLMs. *"Format"* represents the Format Strategy. Although the winning score is only slightly greater than the naive version, this version makes it

possible for LLMs to follow the customized user-defined formats, indicating great potential for the controllability of LLMs. Moreover, the format version can be easily used with other types of meta instructions, showing great extensibility. *"Permute"* represents the Permute Strategy that builds on the Format Strategy with a probability of $1/2$, similar to *"Maskout"*. *"Permute/Maskout"* represents our default setting, where the Permute or Maskout Strategies are utilized together with the Format Strategie with a probability of $1/3$. All these 3 settings show higher performance than the format version, indicating the effectiveness of Advanced Mosaic Strategies which define more complicated meta instructions.

**Ablation on the Max Number of Instructions** is presented in Table 4b, including the pair-wise comparison values. As shown in the table, when the max number is set as 2, i.e. at most 2 instructions/responses are concatenated together, the performance is almost the same as the baseline, indicating the ineffectiveness. However, when the max number grows, the corresponding winning scores also grow consistently. This trend shows that the more instructions concatenated

Table 5: Ablation on the **Distribution of Number of Instructions**. The distribution formula and data counts for different settings are shown in Appendix A. "Mix $\leq 5$" represents the percentage of samples with the number of instructions less or equal to $5$.

|  | Winning Score | Win | Tie | Lose | Mix $\leq 5$ |
|---|---|---|---|---|---|
| Fix | 0.982 | 90 | 34 | 94 | 2.39% |
| Exponential | 0.995 | 94 | 29 | 95 | 2.58% |
| Pareto | 1.417 | 129 | 51 | 38 | 8.94% |
| Log-normal | 1.431 | 136 | 40 | 42 | 6.83% |
| Logistic | 1.417 | 123 | 49 | 46 | 15.84% |
| Uniform | 1.349 | 123 | 48 | 47 | 51.45% |

together, the better the instruction-following ability. We hypothesize that, with the growth of the number of instructions, the overall instruction becomes much harder to follow, especially for the permute and maskout strategies, which benefits LLMs' instruction-following capability.

**Ablation on the Distribution of Number of Instructions** is presented, including the pair-wise comparison values in Table 5 and detailed number distribution comparisons in Figure 3, which aims at identifying how this count distribution affects the performance of our method. The detailed distribution formula and data counts are provided in the Appendix A. *"Fix"* represents the setting where all the overall instructions are concatenated with a fixed number of instructions, which we set as 10 unless the overall

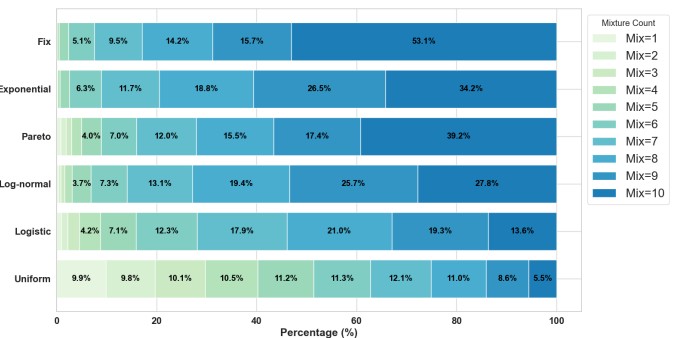

Figure 3: Ablation on the **Distribution of Number of Instructions**, the visualization of distribution comparisons.

instructions exceed the max length limit. *"Exponential"* represents the setting where the number of instructions is sampled following the exponential distribution. Under these two settings, less than 3% of the overall instructions are concatenated by less or equal to 5 original instructions. The lack of few-instruction concatenated samples negatively affects the LLMs' ability to follow the single instruction, which is employed by most of the existing evaluation methods, leading to worse performances. *"Pareto"*, *"Log-normal"*, and *"Logistic"* represents the corresponding distribution that are utilized for sampling. Different from the above two settings, approximately 10% of the overall instructions are composed of fewer original instructions, thus ensuring the LLMs are trained with samples with sufficiently diverse lengths, resulting in optimal performances. *"Uniform"* is our default setting, representing using the uniform distribution where different numbers are sampled evenly. In this situation, the LLMs are trained with samples with the most diverse lengths, thus avoiding the LLMs overfit to simple lengthy responses.

## 5 FURTHER DISCUSSION

### 5.1 PRELIMINARY EXPERIMENTS: PERFORMANCE DEGRADATION

The motivation of our Mosaic-IT is also rooted in the observation that when handling multiple instructions simultaneously, a performance degradation will incurred for even strong LLMs like

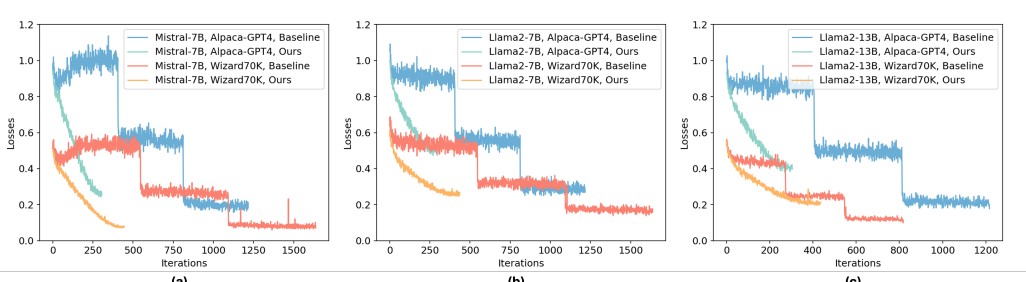

Figure 4: The training loss curve comparisons between the original instruction tuning process and our Mosaic-IT with w datasets on (a) Mistral-7B, (b) Llama2-7B, and (c) Llama2-13B. **The "stair-like" loss curves for the original training process indicate potential memorizing effects, while our loss curves are smoother.** All the training settings are kept the same between the baseline models and Mosaic-IT models, including the Learning Rate, Warm-up Ratio, Learning Rate Schedule (Cosine), Batch Size, etc.

GPT-4-turbo. While LLMs generally perform well when responding to single instructions, their capability to follow multiple instructions at once tends to decline noticeably. BatchPrompt has shown the uncertainty when LLMs are requested to answer multiple formatted questions at one time. Moreover, in some cases, e.g for general open-domain instructions, LLMs might directly ignore some of the instructions, especially when the LLMs are required to respond to the instructions in a random pre-defined order, which is exactly simulating our *Permute* strategy.

To quantitatively analyze this phenomenon, experiments using GPT-3.5-turbo and GPT-4-turbo are conducted on the WizardLM test set. Specifically, we compare the models' performance when responding to multiple instructions concurrently versus responding to a single instruction at each time, by utilizing LLM-based Pair-Wise comparison, as shown in Table 6. All the win rates are lower than 1.0, demonstrating a clear and significant reduction in response quality when these models are required to respond to multiple instructions at one time. Moreover, the possibility of missing instructions (Miss Rate) increases further when they are required to respond to the instructions in a predefined random order rather than a sequential order. These results clearly demonstrate the difficulties of following several instructions at a time and why it can be regarded as a higher level of instruction-following capability.

Table 6: Pair-wise win rate of performances when responding to multiple instructions concurrently versus responding to a single instruction each time, and miss rate when responding to multiple instructions concurrently. "3 Instructions" represents the setting where 3 random instructions are concatenated together for inference. "Sequential" and "Random" represents the setting where the models are asked to respond to each instruction sequentially, or in a random pre-defined order.

| Pair-Wise (Multi vs. Single) | 3 Instructions | | 5 Instructions | | 7 Instructions | |
|---|---|---|---|---|---|---|
| | Win Rate ↑ | Miss Rate ↓ | Win Rate ↑ | Miss Rate ↓ | Win Rate ↑ | Miss Rate ↓ |
| GPT-3.5-turbo (Sequential) | 0.357 | 0.014 | 0.336 | 0.055 | 0.303 | 0.064 |
| GPT-3.5-turbo (Random) | 0.315 | 0.124 | 0.330 | 0.156 | 0.198 | 0.312 |
| GPT-4-turbo (Sequential) | 0.176 | 0.000 | 0.137 | 0.000 | 0.140 | 0.000 |
| GPT-4-turbo (Random) | 0.139 | 0.000 | 0.153 | 0.014 | 0.101 | 0.005 |

## 5.2 MOSAIC: ALLEVIATING MEMORIZING

In the original instruction tuning process, each data sample will be trained several times for LLMs without changes to the instructions and responses. This training process poses risks to the potential memorizing effects on training samples, which can be partially indicated by the "stair-like" training loss curves as shown in Figure 4. In the figure, all the training settings are kept the same between the baseline models and Mosaic-IT models, including the Learning Rate, Warm-up Ratio, Learning Rate Schedule (Cosine), Batch Size, etc. For the baseline methods, the training loss hardly decreases within each epoch of training but drops dramatically when the LLMs meet the same training samples again, which indicates a potential memorizing effect of training samples and potential overfitting. However, when utilizing our method, the random mosaics of original instructions with diverse

Table 7: The training time comparison of different settings, and the pair-wise winning scores are also provided for better illustration. "Uni-2" represents uniform distribution with max count as 2. **Mosaic-IT reduces the training time to** $16\% - 25\%$ **while achieving better performance.**

| Settings | Baseline | Fix | Exponential | Pareto | Log-normal | Logistic | Uni-2 | Uni-4 | Uni-6 | Uni-8 | Uni-10 | Uni-12 |
|---|---|---|---|---|---|---|---|---|---|---|---|---|
| Time (min) | 827 | 121 | 129 | 133 | 133 | 143 | 716 | 426 | 305 | 245 | 202 | 173 |
| Time Ratio | 100.0% | 14.6% | 15.6% | 16.1% | 16.1% | 17.3% | 86.6% | 51.5% | 36.9% | 29.6% | 24.4% | 20.9% |
| Winning Score | 1.000 | 0.982 | 0.995 | 1.417 | 1.431 | 1.417 | 0.989 | 1.142 | 1.303 | 1.294 | 1.349 | 1.376 |

and complex meta-instructions largely diversify the overall training instructions. Although each original data sample will still be seen by LLMs several times during training, the overall context varies dramatically as each original sample is only an atomic element of the overall mosaic sample, indicating that there will be no identical overall instructions during the whole training process. Thus this augmentation largely alleviates the potential memorizing and overfitting problems as shown in the figure, where the training loss decreases smoothly, representing the gradual learning process.

## 5.3 MOSAIC: IMPROVING EFFICIENCY

One of the benefits of our method is the efficiency of the training process. Given an existing dataset, our mosaic processes largely decrease the number of total overall instructions and the total number of gradient descents, leading to a reduction in the training process. The detailed comparison is shown in Table 7, which is based on the Mistral-7B model on the Alpaca-GPT4 dataset. The time is calculated based on four NVIDIA A100 Graphic Cards. As shown, our method greatly decreases the training time to approximately $16\%$ to $25\%$ while achieving better performances, especially when there are mosaic samples with larger permutation counts.

## 5.4 MOSAIC: WHY IT WORKS?

The effect of our method is aligned with the Dense and Aligned Captions Doveh et al. (2023) used in VL Models, which utilizes denser captions to promote the VL models. Different from all previous methods which require LLMs to generate the whole response conditioned on the whole instruction, our method forces LLMs to generate responses selectively conditioned on some critical parts of the overall dense instruction. Especially in advanced strategies, LLMs are required to generate responses conditioned not only on sequential parts of the instruction, but diverse and randomized segments of it, which is defined by meta-instruction.

Compared to original settings, our setting requires LLMs to first understand the formats and orders defined in the meta-instructions, then adapt to conditioning on different parts of the instructions when generating responses. This process forces LLMs to develop a more comprehensive understanding of context, prioritize various pieces of information, and manage complex dependencies between instructions, thus improving instruction-following performance. Moreover, the entire process is data- and model-agnostic, ensuring the generalizability of our method.

## 5.5 LIMITATION AND FUTURE DIRECTIONS

The potential limitations of our work: (1) Currently, three Advanced Mosaic Strategies with corresponding high-level rules are proposed and utilized in our method, however, we believe more strategies and predefined rules can be further introduced. (2) The optimal distribution of the number of instructions for the mosaic process still needs further justification in future studies. (3) It is unknown whether the inclusion of extra models or careful curation/selection of instructions for concatenation will further improve the performance of Mosaic-IT largely.

## 6 CONCLUSION

We introduce Mosaic Instruction Tuning (Mosaic-IT), a novel, human/model-free method to enhance instruction tuning for LLMs. By concatenating multiple instruction-response samples and using higher-level meta-instructions, Mosaic-IT improves multi-step and format-following capabilities. Our evaluations show superior performance and an $80\%$ reduction in training costs compared to the original methods. Mosaic-IT's simplicity and efficiency make it a scalable solution for improving LLMs without extensive human intervention or resource-intensive teacher models. Our results highlight the potential of innovative data augmentation techniques in advancing LLM capabilities.

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

## A    DETAILED DISTRIBUTION FOR ABLATION ON MIXTURE DISTRIBUTION

### A.1    DISTRIBUTION DESCRIPTION

The detailed distribution descriptions and formulas are provided below.

**Exponential Distribution**[1]: The exponential distribution is a continuous probability distribution used to model the time or space between events in a Poisson process. The probability density function (PDF) of the exponential distribution is:

$$f(x; \lambda) = \lambda e^{-\lambda x} \quad \text{for } x \geq 0,$$

where $\lambda = 1$ by default in our setting. We will resample with this distribution if the sampled value $x_{sample}$ is greater then $k_{max}$.

**Log-normal Distribution**[2]: The log-normal distribution is a continuous probability distribution of a random variable whose logarithm is normally distributed. It is often used to model variables that are positively skewed, such as income, stock prices, and other financial data. The probability density function (PDF) for a log-normal distribution is given by:

$$f(x; \mu, \sigma) = \frac{1}{x\sigma\sqrt{2\pi}} \exp\left(-\frac{(\ln x - \mu)^2}{2\sigma^2}\right) \quad \text{for} \quad x > 0$$

where $\mu = 0$ and $\sigma = 0$ by default in our setting. We will resample with this distribution if the sampled value $x_{sample}$ is greater than $k_{max}$.

**Logistic Distribution**[3]: The logistic distribution is a continuous probability distribution used in various fields, including logistic regression, modeling growth, and in some cases as an alternative to the normal distribution due to its heavier tails. The probability density function (PDF) for the logistic distribution is given by:

$$f(x; \mu, s) = \frac{e^{-(x-\mu)/s}}{s\left(1 + e^{-(x-\mu)/s}\right)^2}$$

where $\mu = 0$ and $s = 2$ by default in our setting. We will resample with this distribution if the sampled value $x_{sample}$ is greater than $k_{max}$.

**Pareto Distribution**[4]: The Pareto II or Lomax distribution is a shifted Pareto distribution. It can be considered as a simplified version of the Generalized Pareto distribution, with the scale set to one and the location set to zero. The probability density function (PDF) for the Pareto distribution is:

$$f(x; \alpha) = \frac{\alpha m^\alpha}{x^{\alpha+1}} \quad \text{for} \quad x \geq m,$$

where $m = 1$ and $\alpha = 1$ by default in our setting. We will resample with this distribution if the sampled value $x_{sample} - 1$ is greater than $k_{max}$.

After getting $x_{sample}$, a floor function will be utilized to get the corresponding integer and the final concatenation count $k = k_{max} - floor(x_{sample})$.

### A.2    DISTRIBUTION VISUALIZATION

The detailed data counts for different distributions are provided in Figure 5.

---

[1]https://numpy.org/doc/stable/reference/random/generated/numpy.random.exponential.html

[2]https://numpy.org/doc/stable/reference/random/generated/numpy.random.lognormal.html

[3]https://numpy.org/doc/stable/reference/random/generated/numpy.random.logistic.html

[4]https://numpy.org/doc/stable/reference/random/generated/numpy.random.pareto.html

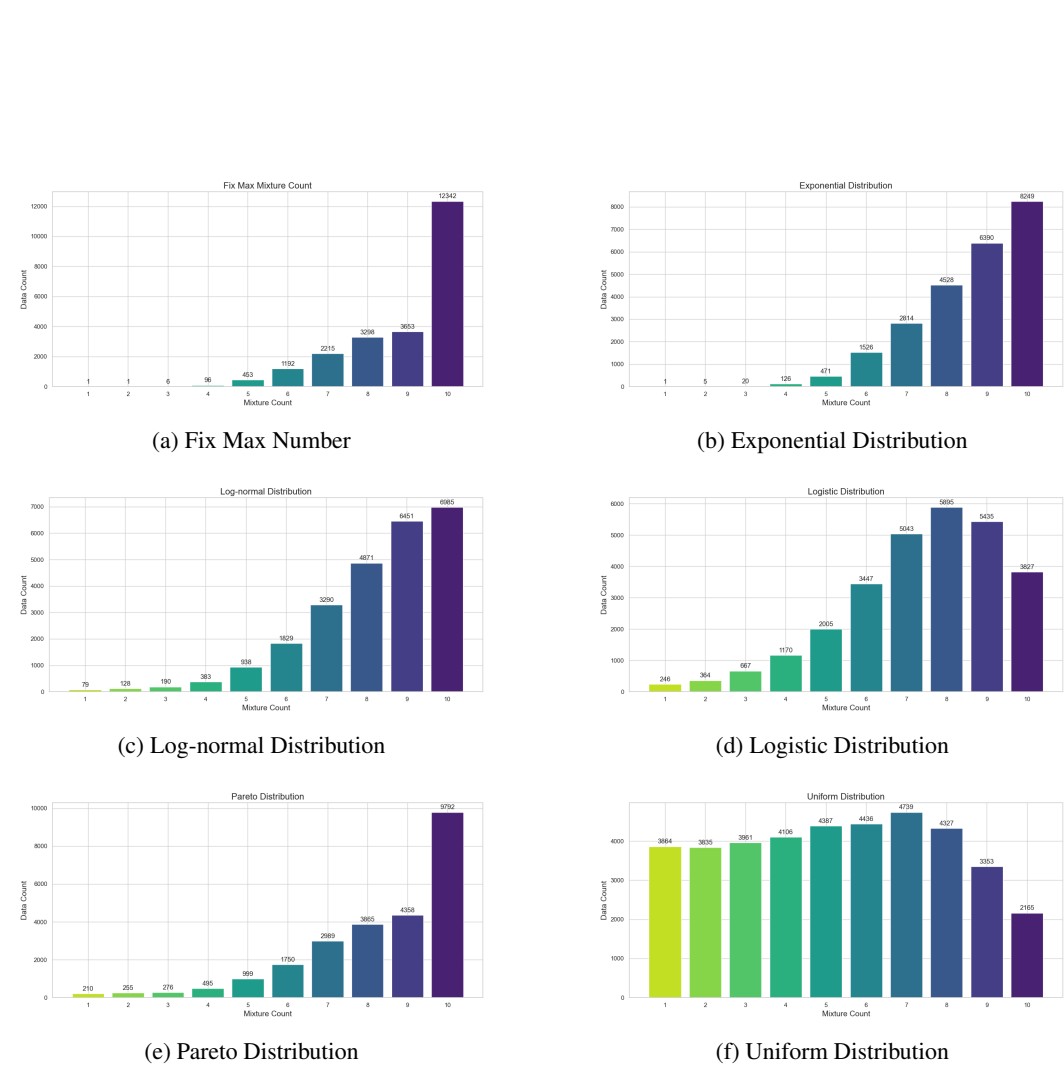

(a) Fix Max Number

(b) Exponential Distribution

(c) Log-normal Distribution

(d) Logistic Distribution

(e) Pareto Distribution

(f) Uniform Distribution

Figure 5: Bar plots of detailed data counts for different distributions in the Ablation on the Numbers of Instructions: (a) Fix Max Number, (b) Exponential Distribution, (c) Log-normal Distribution, (d) Logistic Distribution, (e) Pareto Distribution, (f) Uniform Distribution.

## B  EXPERIMENTAL SETUP

### B.1  IMPLEMENTATION DETAILS

For the three base pre-trained models, Llama2-7B, Llama2-13B Touvron et al. (2023b), and Mistral-7B Jiang et al. (2023), we utilize the prompt and code base from Vicuna Chiang et al. (2023) and flash attention Dao et al. (2022). The overall training arguments are aligned with the common training configuration. The Adam optimizer Kingma & Ba (2017) is utilized with the batch size of $128$ and with the max token length of $2048$. When training the baseline models Llama2-7B and Llama2-13B, the maximum learning rate is set to $2 \times 10^{-5}$ with the warmup rate as $0.03$ for 3 epochs. When training the baseline models on Mistral-7B, the maximum learning rate is set to $1 \times 10^{-5}$ with the warmup rate as $0.1$ for 3 epochs. For the three models, Llama-3-8B Dubey et al. (2024), Phi-3 Abdin et al. (2024), and Gemma2-2B Team et al. (2024), we utilize the code base from LLaMA-Factory Zheng et al. (2024b). The max token length is set with $4096$ following the modern settings and we train the model for 2 epochs. Other parameters are kept the same as the above.

When training with Mosaic-IT, we run the mosaic process $n$ times for each experiment to simulate $n$ epochs of training, $n$ represents the number of epochs trained on baseline models, to ensure the alignment of overall data sample counts. Then these augmented data are mixed together and used for training 1 epoch while all other configurations are kept the same as baselines.

### B.2  TRAINING DATASET

The Alpaca dataset Taori et al. (2023) comprises $52,000$ instruction-following samples and is constructed utilizing the self-instruct paradigm Wang et al. (2023b). This dataset was produced by employing OpenAI's text-davinci-003 model. Characterized as a classical dataset with moderate quality attributes, the Alpaca dataset serves as an initial platform to validate our methodology. To further substantiate our approach using a dataset of inherently high quality, we also applied our method to the Alpaca-GPT4 dataset Peng et al. (2023), which features responses generated by GPT4. The WizardLM dataset Xu et al. (2023) is also utilized in our method, which contains $70,000$ samples created by the evolution algorithm proposed by them. With ChatGPT-3.5 utilized, the data quality on WizardLM is largely guaranteed. The Vicuna 1M dataset Zheng et al. (2024a) is a large-scale dataset containing one million real-world conversations with 25 state-of-the-art LLMs, due to the computation budget, 300k instances are randomly sampled for our experiments. Magpie dataset Xu et al. (2024b) is a most recent SOTA synthetic dataset with 300k samples.

### B.3  EVALUATION METRICS

**Pair-wise Comparison** by using powerful LLMs like GPT-4 is recently widely accepted and becoming a common practice Touvron et al. (2023b); Chiang et al. (2023); Dettmers et al. (2023); Liu et al. (2023b); Chiang & Lee (2023). The evaluation of responses from LLMs, especially in open-domain contexts where definitive ground truth is hard to establish, continues to be an intricate and evolving research domain. Recent studies, however, have indicated a notable alignment between GPT-4's performance evaluations and human assessments Zheng et al. (2023); Li et al. (2023c); Sottana et al. (2023), thereby establishing a credible foundation for this evaluative methodology. We adopted test instruction sets from WizardLM Xu et al. (2023), comprising 218 diverse, human-curated instructions for pair-wise comparison. We directly follow the evaluation framework proposed by Chen et al. (2023); Li et al. (2024d), which evaluates responses on a scale spanning from 1 to 10 across multiple dimensions. To further address positional bias, as discussed by Ko et al. (2020); Wang et al. (2023a), the comparison is conducted in two distinct sequences, LLM1's response first and then LLM2's response first, ensuring a fair assessment of model performance. Evaluation outcomes are categorized into 'win-tie-loss' for each instruction. The detailed evaluation prompt is provided in Appendix E.

**Open LLM Leaderboard**, employing the evaluation framework from Eval Harness Gao et al. (2021), offers a detailed and systematic approach to assessing the capabilities of generative language models through a set of diverse evaluation tasks. This methodology zeroes in on four pivotal benchmarks: ARC Clark et al. (2018), HellaSwag Zellers et al. (2019), MMLU Hendrycks et al. (2021), and TruthfulQA Lin et al. (2022). These benchmarks collectively provide a comprehensive

evaluation of the models' reasoning abilities, their grasp of common-sense knowledge, and their accuracy in presenting factual information. Consequently, the leaderboard presents valuable insights.

**Alpaca-Eval Leaderboard**, leveraging the AlpacaFarm evaluation dataset, presents a dependable and efficient automated evaluation tool for LLMs Li et al. (2023c); Dubois et al. (2023). This tool benchmarks the responses generated by LLMs against those from Davinci003, focusing on the models' ability to adhere to generic user instructions.

**MT-Bench** (Multi-turn Benchmark) Zheng et al. (2023) is a benchmark tool designed for automated evaluating LLMs in multi-turn dialogue settings. It focuses on analyzing conversation flow and the model's ability to follow instructions with 80 high-quality, multi-turn questions.

**IFEval** (Instruction-Following Eval) Zeng et al. (2024) is a straightforward and easy-to-produce evaluation benchmark focusing on a set of "verifiable instructions". It contains 25 types of verifiable instructions and 541 prompts, with each prompt containing one or multiple verifiable instructions.

## C  RELATED WORK

Earlier research in instruction tuning primarily centered on constructing expansive, high-quality datasets through intensive curation by human experts, a process both time-consuming and labor-intensive Khashabi et al. (2020); Ye et al. (2021); Wei et al. (2022); Wang et al. (2022); Du et al. (2022). Motivated by the success of Alpaca Taori et al. (2023), recent studies have explored automated approaches for developing instruction-tuning datasets.

**Instruction Data Improvement:**  WizardLM Xu et al. (2023) first proposes an Evol Algorithm to complicate the existing data and reach supreme performance. LaMini-LM Wu et al. (2024) innovatively generates "Topic-Guided" instructions utilizing Wiki data. Tree-Instruct Zhao et al. (2024) preliminarily explores the relationship between instruction complexity and Alignment and proposes adding nodes to complicate the instruction. UltraChat Ding et al. (2023) establishes broad thematic scopes, systematically generating numerous instructions within each. Reflection-Tuning Li et al. (2023a) sequentially refines both instructions and responses by focusing on specific evaluative criteria. DEITA Liu et al. (2023a) utilizes ChatGPT to diversify and then select the data. Selective Reflection-Tuning Li et al. (2024b) proposes a teacher-student collaborative pipeline to improve and select the data. Instruction Fusion Guo et al. (2024) proposes to utilize ChatGPT4 to merge two distinct instructions for further complexity enhancement. These advancements showcase a shift towards automating the generation and refinement of datasets, reducing reliance on human labor.

**Instruction Data Selection:**   It is widely accepted that "quality is all you need" Touvron et al. (2023b); Zhou et al. (2023) for instruction tuning. LIMA Zhou et al. (2023) demonstrates that merely 1,000 human-carefully-curated, high-quality training instances can substantially enhance the instruction-following performance. InsTag Lu et al. (2023) employs the proprietary model, ChatGPT, to tag instruction data and select data with complex tags. Alpagasus Chen et al. (2023) utilizes proprietary LLMs chatGPT and Claude2 to directly assess the quality of instruction tuning data. Cherry LLM Li et al. (2024d) proposes the Instruction-Following Difficulty (IFD) scores to assess the difficulty of the instructions, which is a self-guided method in which no extra LLMs are utilized. Motivated by Humpback Li et al. (2023b), Selective Reflection-Tuning Li et al. (2024b) extends the IFD score to a reverse version, focusing on the feasibility of responses. Du et al. (2023) and Bukharin & Zhao (2023) utilize reward models as the base scores for measuring data quality. DEITA Liu et al. (2023a) experiments on several different data selection metrics and builds a dataset with high quality. Superfiltering Li et al. (2024c) reveals the consistency between weak and strong language models in perceiving instruction difficulty, making the filtering process much more efficient. All these works are devoted to distinguishing and selecting good data samples from bad ones for instruction tuning.

# D PREDEFINED RULES

Examples of predefined formats can be found in Table 8 and detailed predefined rule descriptions can be found in Table 9.

Table 8: Examples of predefined formats, including the Serial Digit formats and Response Parsing formats. "*i*" represents the real number serial number, "*text*" represents the replaceable parsing text, and "*response*" represents the real response of the concatenated overall instructions/responses. The response parsing formats are composed of the parsing bracket and text. In each mosaic process, random formats will be sampled simulating the real-world user-defined formats. The last column represents the assembled examples using the formats in the same row.

| Serial Digit | Parsing Bracket | Parsing Text | Assembled Examples |
|---|---|---|---|
| *i* | (*text*) | BEGIN, END | 1. (BEGIN)*response*(END) |
| (*i*) | [*text*] | START, END | (1). [START]*response*[END] |
| [*i*] | ⟨*text*⟩ | RESPONSE, END | [1]. ⟨RESPONSE⟩*response*⟨END⟩ |
| ⟨*i*⟩ | ≪*text*≫ | RESPONSE, END OF RESPONSE | ⟨1⟩. ≪RESPONSE≫*response*≪END OF RESPONSE≫ |
| ≪*i*≫ | \|*text*\| | OPEN, CLOSE | ≪1≫. \|OPEN\|*response*\|CLOSE\| |
| ###*i* | [\|*text*\|] | OPEN RESPONSE, CLOSE | ###1. [\|OPEN RESPONSE\|]*response*[\|CLOSE\|] |
| ##*i* | ⟨\|*text*\|⟩ | INITIATE, TERMINATE | ##1. ⟨\|INITIATE\|⟩*response*⟨\|TERMINATE\|⟩ |
| ##*i*## | #*text*# | START POINT, END POINT | ##1##. #START POINT#*response*#END POINT# |
| \|*i*\| | *\*text\** | RES_START, RES_END | \|1\|. *RES_START**response**RES_END* |
| \|\|*i*\|\| | @*text*@ | RES, /RES | \|\|1\|\|. @RES@*response*@/RES@ |

Table 9: Predefined rules for the Permute and Maskout strategy. A random rule will be sampled for each mosaic process, which largely complicates and diversifies the mosaicked instructions.

| Strategy | Rule Name | Rule Description |
|---|---|---|
| Permute | FIX | Respond in the order of a provided list. |
| Permute | REVERSE | Respond in reverse of the original order. |
| Permute | ALPHA | Respond in the alphabetical order of the first letter of tasks. |
| Permute | REVERSE_ALPHA | Respond in the reverse alphabetical order of the first letter of tasks. |
| Permute | LENGTH_WORD | Respond according to the length (words) of tasks, respond to short ones first. |
| Permute | REVERSE_LENGTH_WORD | Respond according to the length (words) of tasks, respond to long ones first. |
| Permute | LENGTH_CHAR | Respond according to the length (characters) of tasks, respond to short ones first. |
| Permute | REVERSE_CHAR_WORD | Respond according to the length (characters) of tasks, respond to long ones first. |
| Permute | ODD_EVEN | First respond to the odd-numbered tasks, then the even-numbered ones. |
| Permute | EVEN_ODD | First respond to the even-numbered tasks, then the odd-numbered ones. |
| Maskout | FIX | Ignore the tasks provided in the list. |
| Maskout | WORD_LONG | Ignore the longest one/several task(s) according to the word count. |
| Maskout | WORD_SHORT | Ignore the shortest one/several task(s) according to the word count. |
| Maskout | ODD | Ignore the odd-numbered tasks. |
| Maskout | EVEN | Ignore the even-numbered tasks. |

# E  PROMPT FOR EVALUATION

The detailed pair-wise comparison prompt for the pair-wise comparison is in Figure 6.

---

Prompt for Performance Evaluation

---

**System Prompt**
You are a helpful and precise assistant for checking the quality of the answer.

**User Prompt**
[Question]
*Question*
[The Start of Assistant 2's Answer]
*Answer 2*
[The End of Assistant 2's Answer]
[The Start of Assistant 2's Answer]
*Answer 2*
[The End of Assistant 2's Answer]

We would like to request your feedback on the performance of two AI assistants in response to the user question displayed above.
Please rate the helpfulness, relevance, accuracy, level of details of their responses. Each assistant receives an overall score on a scale of 1 to 10, where a higher score indicates better overall performance.
Please first output a single line containing only two values indicating the scores for Assistant 1 and 2, respectively. The two scores are separated by a space. In the subsequent line, please provide a comprehensive explanation of your evaluation, avoiding any potential bias and ensuring that the order in which the responses were presented does not affect your judgment.

---

Figure 6: The prompt we used to request GPT4-Turbo to evaluate the responses.

# F   DETAILED PERFORMANCE SCORES ON LLAMA3, PHI3 AND GEMMA2

The detailed performance scores on the Open LLM Leaderboard and IFEval, for Llama-3-8B, Phi-3, and Gemma2-2B.

Table 10: The performance comparison on more model families and datasets on all five automatic evaluation metrics. In IF Eval, P and I represent Prompt-level and Instruction-level accuracy.

| Model | Dataset | Method | Open LLM Leaderboard ↑ | | | | | IF Eval ↑ | | | |
|---|---|---|---|---|---|---|---|---|---|---|---|
| | | | Average | ARC | HellaSwag | MMLU | TruthfulQA | Prompt (S) | Inst (S) | Prompt (L) | Inst (L) |
| **Llama-3-8B** | Vicuna | Baseline | 52.51 | 44.54 | 70.66 | 49.68 | 45.18 | 19.04 | 30.70 | 21.26 | 33.45 |
| | | Mosaic-IT | **55.62** | 47.78 | 73.77 | 56.11 | 44.83 | **29.76** | **43.17** | **31.42** | **45.56** |
| | Magpie | Baseline | 56.15 | 50.09 | 71.29 | 54.40 | 48.84 | 29.39 | 40.76 | 35.67 | 47.72 |
| | | Mosaic-IT | **60.13** | 53.58 | 76.62 | 60.82 | 49.52 | **38.08** | **49.64** | **40.67** | **52.76** |
| **Phi-3** | Vicuna | Baseline | 62.06 | 58.96 | 76.48 | 64.89 | 47.89 | 28.47 | **40.29** | 30.50 | **43.17** |
| | | Mosaic-IT | **62.30** | 58.45 | 77.66 | 65.24 | 47.87 | **30.13** | 39.57 | **32.35** | 41.85 |
| | Magpie | Baseline | 62.90 | 59.30 | 75.07 | 65.89 | 51.35 | 39.56 | 50.84 | 44.36 | 55.25 |
| | | Mosaic-IT | **63.54** | 60.23 | 76.30 | 66.14 | 51.50 | **42.33** | **53.60** | **50.83** | **62.35** |
| **Gemma2-2B** | Vicuna | Baseline | 48.90 | 43.43 | 64.20 | 41.50 | 46.46 | 20.51 | 32.61 | 23.66 | 35.61 |
| | | Mosaic-IT | **51.31** | 46.33 | 69.32 | 44.29 | 45.31 | **21.44** | **33.57** | **24.03** | **36.93** |
| | Magpie | Baseline | 46.37 | 39.59 | 60.71 | 35.46 | 49.75 | 19.78 | 29.74 | 21.81 | 32.49 |
| | | Mosaic-IT | **48.36** | 39.33 | 64.10 | 39.87 | 50.16 | 19.78 | **31.65** | **22.18** | **34.77** |

