# OpenReview forum: "Mosaic-IT: Free Compositional Data Augmentation Improves Instruction Tuning"
_ICLR.cc/2025/Conference — Submitted to ICLR 2025_

### Official Review · Reviewer_7WNG · 2024-10-26

**Soundness:** 2
**Presentation:** 2
**Contribution:** 2
**Rating:** 5
**Confidence:** 4

**Summary:**

This paper proposes Mosaic-IT, a data augmentation method for the instruction following. It proposes Primary and advanced mosaic strategies. It also includes format, permute, and mark out.

**Strengths:**

The paper deploys the method into several evaluation benchmarks and different model structures. It also includes several analyses to study the method's aspects.

**Weaknesses:**

1.	The core problem of the proposed method is the lack of a detailed explanation of the core reasons for the proposed method. It does not provide justifiable and experimental explanations for the effectiveness of the proposed method. The primary motivation of the proposed method should be further cleared here.
2.	From the experiments, it seems that the proposed method does not improve the multi-turn data for MT-Bench. Would there be any explanations for this?
3.	Many instruction-following methods and literatures focus on data augmentation. There are no comparisons with those baselines. In addition, how would mask methods be different from the other dropout, etc. methods?

**Questions:**

1. Would there be any justifiable and experimental explanations for demonstrating the method's effectiveness?

2. Would it be possible to show more experiments on multi-turn benchmarks?

3. For the comparison baselines, would it be possible to add more baselines related to data augmentation for instruction turning?

---

> ### Author Response · Authors · 2024-11-24
> **Response to Reviewer #4(7WNG)**
>
> **Weakness:**
>
> >Q1: The core problem of the proposed method is the lack of a detailed explanation of the core reasons for the proposed method. It does not provide justifiable and experimental explanations for the effectiveness of the proposed method. The primary motivation of the proposed method should be further cleared here.
>
> Please kindly refer to the Q1 of the General response.
>
> >Q2: From the experiments, it seems that the proposed method does not improve the multi-turn data for MT-Bench. Would there be any explanations for this?
>
> 1. Mosaic-IT does not necessarily improve the second-round dialogue of LLMs if the instruction data are single-round conversations. Our method composites several instructions into one but it is still under the setting of single round conversations.
> 2. The second-round performances are affected by both the instruction data and the LLMs to be trained. As shown in Table 3, when more advanced data and LLMs are used, the second-round performances can be further improved.
>
> More discussion will be included in the later version.
>
> >Q3.1: Many instruction-following methods and literatures focus on data augmentation. There are no comparisons with those baselines.
>
> Most existing instruction-following methods and literature focus on data augmentation **utilizing other LLMs to generate new data samples**, which is a data synthesis process relying on other LLMs. However, **our method is a model-free method that aims to exploit the potential of existing data**, which is largely different from existing methods. To our best knowledge, our method is the first of this kind. Please let us know if you find any works of this kind. In more detail:
>
> 1. Our method avoids the entire cost of data synthesis by any language models or neural networks, which still produce carbon footprints.
> 2. Our method aims to **exploit the potential of existing data** so the quality of augmented data is 100% guaranteed. In contrast, the newly generated data still require careful and complex quality checks in practice to reduce the risk of degrading the original LLM (i.e., negative finetuning).
> 3. **Mosaic-IT is orthogonal and complementary to existing data synthesis approaches**. It can be applied together with other methods to further improve LLMs.
> 4. As a model-free method, Mosaic-IT can avoid **the potential violation of licenses** limiting the usage of existing LLMs.
>
>
>
> >Q3.2: In addition, how would mask methods be different from the other dropout, etc. methods?
>
> There might be some misunderstanding of the maskout strategy, which is applied to the training data to create data augmentations with different outputs. This is different from dropout-typed approaches that are applied to neurons or parameters of a model, whose goal is to perturb and regularize the neural network training. But it does not change the output in the training data.
>
> By maskout in Mosaic-IT, we instruct LLMs to ignore some of the instructions and when generating the responses and train LLMs to follow such meta-instructions. We will consider changing the name of this strategy to avoid any misunderstanding.

---

> ### Author Response · Authors · 2024-11-25
> **A kind reminder**
>
> Dear Reviewer 7WNG,
>
> As we are approaching the deadline of the discussion period, we would like to cordially inquire about the extent to which we have successfully addressed the concerns outlined in your review. Should there be any lingering points that require further attention, please rest assured that we are enthusiastic about the opportunity to provide comprehensive responses to any subsequent queries or comments you may have.
>
> Your constructive input remains invaluable to us, and we appreciate your dedication to enhancing the quality of our manuscript. Thank you for your time and consideration.
>
> Best,
>
> Authors

---

> ### Author Response · Authors · 2024-11-27
>
> Thanks for the prompt response! We are still confused about how to apply image augmentations MixMatch or MixUp for representation learning to text data for autoregressive language modeling (in particular, finetuning of modern LLMs such as Llama, which is our focused problem). To the best of our knowledge, whether and how these methods can be applied to LLM finetuning is still unknown as both models and the training objectives are very different. We are eager to hear from you how these methods can be leveraged for the LLM finetuning. Thanks!

---

> > ### Author Response · Authors · 2024-11-29
> > **A further kin reminder**
> >
> > Dear Reviewer 7WNG,
> >
> > As we are approaching the deadline of the discussion period, we would like to cordially inquire about the extent to which we have successfully addressed the concerns outlined in your review. Please kindly let us know if you have any further concerns.
> >
> > Your constructive input remains invaluable to us, and we appreciate your dedication to enhancing the quality of our manuscript. Thank you for your time and consideration. If our response addresses your concerns, we sincerely hope you can consider raising the ratings. Thank you so much!
> >
> > Best,
> >
> > Authors

---

> > > ### Author Response · Authors · 2024-12-03
> > > **A further kind reminder**
> > >
> > > Dear Reviewer 7WNG,
> > >
> > > Since the discussion period is about to end soon, we would like to cordially inquire about the extent to which we have successfully addressed the concerns outlined in your review.
> > >
> > > Your constructive input remains invaluable to us, and we appreciate your dedication to enhancing the quality of our manuscript. Thank you for your time and consideration. If our response addresses your concerns, we sincerely hope you can consider raising the ratings. Thank you so much!
> > >
> > > Best,
> > >
> > > Authors

---

### Official Review · Reviewer_cPom · 2024-11-03

**Soundness:** 3
**Presentation:** 3
**Contribution:** 3
**Rating:** 6
**Confidence:** 5

**Summary:**

This paper introduces a data augmentation method, Mosaic-IT, for instruction-tuning large language models (LLMs) without human or model dependency. Unlike traditional approaches that rely on human intervention or teacher models to generate instruction-response pairs, the proposed method works by combining existing instructions into composite multi-instruction samples. They propose four ways to do the composition - primary, format, permute and maskout. By doing so, the paper shows that LLMs trained with this method develop a higher level of instruction-following capacity and format adherence. The proposed method, which reduces training time by approximately 80%, holds promise as a scalable solution for instruction tuning without extensive resource requirements.

**Strengths:**

1. The paper is well-structured, progressing logically from the motivation behind Mosaic-IT to the methodology, followed by experiments and results. Each section builds on the last, making the paper easy to follow and understand.
2. The figures do a great job of clearly summarizing the idea.
3. The experiments are comprehensive for the scope the paper setup - they have explored different datasets and model families and explored different sampling procedures for the composition

**Weaknesses:**

1. The paper lacks a theoretical basis for why random concatenation should improve instruction-following abilities; structured or semantically grouped concatenations could offer further insights.
2. Randomly concatenated instructions may introduce noise, potentially impacting training stability. An analysis of this effect on model perplexity would strengthen the work.

**Questions:**

Please refer to the weaknesses

---

> ### Author Response · Authors · 2024-11-24
> **Response to Reviewer #3(cPom)**
>
> **Weakness**:
>
> >Q1: The paper lacks a theoretical basis for why random concatenation should improve instruction-following abilities; structured or semantically grouped concatenations could offer further insights.
>
> Please kindly refer to the Q1 of the General response.
>
> >Q2: Randomly concatenated instructions may introduce noise, potentially impacting training stability. An analysis of this effect on model perplexity would strengthen the work.
>
> Thank you for your insightful suggestions!
>
> We examined the training stability by checking the training curves of loss on different LLMs. Similar to the curves in Figure 4, the starting loss of our method is typically higher compared to the baseline training curves due to the difficulty of the compositional data. This indicates the weakness of existing LLMs in following these compositional instructions, possibly due to the noises or the interference among multiple instructions. However, during the training phase, the loss declines consistently and stably, which indicates that the LLM makes progress on generating each response selectively or in a predefined order according to the corresponding instruction and meta-instruction, without being affected by the interference or noise from other instructions. More discussion will be included in our later version.

---

> ### Author Response · Authors · 2024-11-25
> **A kind reminder**
>
> Dear Reviewer cPom,
>
> As we are approaching the deadline of the discussion period, we would like to cordially inquire about the extent to which we have successfully addressed the concerns outlined in your review. Should there be any lingering points that require further attention, please rest assured that we are enthusiastic about the opportunity to provide comprehensive responses to any subsequent queries or comments you may have.
>
> Your constructive input remains invaluable to us, and we appreciate your dedication to enhancing the quality of our manuscript. Thank you for your time and consideration.
>
> Best,
>
> Authors

---

> > ### Author Response · Authors · 2024-11-29
> > **A kind reminder**
> >
> > Dear Reviewer cPom,
> >
> > As we are approaching the deadline of the discussion period, we would like to cordially inquire about the extent to which we have successfully addressed the concerns outlined in your review. Please kindly let us know if you have any further concerns.
> >
> > Your constructive input remains invaluable to us, and we appreciate your dedication to enhancing the quality of our manuscript. Thank you for your time and consideration. If our response addresses your concerns, we sincerely hope you can consider raising the ratings. Thank you so much!
> >
> > Best,
> >
> > Authors

---

> > > ### Comment · Reviewer_cPom · 2024-12-01
> > > **Response to authors**
> > >
> > > Thank you for your response. While your explanation highlights how Mosaic-IT trains LLMs to handle compositional reasoning through meta-instructions, it does not fully address why random concatenation is the optimal strategy. If following meta-instructions improves compositional reasoning, grouping instructions by semantics, topic, or structure could plausibly enhance this effect by reducing noise and providing more meaningful learning signals. To strengthen your argument, it would be helpful to include empirical comparisons between random concatenation and structured or semantically grouped approaches in the future revision. That said, my current rating reflects my overall assessment of the work, and I believe the paper’s contributions are valuable despite these areas for improvement.

---

> > > > ### Author Response · Authors · 2024-12-03
> > > > **Further experiments and responses to Reviewer cPom**
> > > >
> > > > Thanks for your follow-up response! We would like to address your remaining concern about whether random concatenation is the best strategy for our proposed compositional augmentations.
> > > >
> > > > >Q1: It does not fully address why random concatenation is the optimal strategy.
> > > >
> > > > We did **not claim that random concatenation is the optimal strategy**. We chose it because it demonstrates that: **as the first attempt of model-free compositional augmentation for LLMs, the most straightforward random concatenation already brings significant improvement**. Compared with more sophisticated concatenation strategies, random concatenation stands out as it is **completely cost-free** and does not require any prior knowledge or semantic understanding of the concatenated samples. This supreme efficiency represents one of our greatest contributions. We discussed this in the Limitation section of our manuscript. We will extend this discussion in our future version.
> > > >
> > > > >Q2: To strengthen your argument, it would be helpful to include empirical comparisons between random concatenation and structured or semantically grouped approaches in the future revision.
> > > >
> > > > We agreed that including other concatenation methods could further strengthen our argument. So we follow your suggestion by further conducting experiments of a semantic grouping approach based on the Alpaca-GPT4 dataset to finetuning Mistral. Due to the limited time, we cannot fully extend the experiments to more datasets and models. But we will add them in our future versions.
> > > >
> > > > **Semantic Grouping:**
> > > > We utilized “sentence-transformers/all-mpnet-base-v2” to obtain the semantic embedding for each sample in the dataset, and then we applied the K-means algorithm to group these data samples into multiple clusters. To ensure enough samples per cluster, we set K=52 as the dataset contains 52k samples. Given the clusters, each concatenated sample is composed of multiple samples randomly drawn from the same cluster. We keep using the same training hyperparameters as before. In the table below, we report the performance on 2 evaluation metrics: pair-wise comparison and Alpaca Eval.
> > > >
> > > > | Method | Alpaca Eval 2 (LC) |  Alpaca Eval 2  | Pair-wise Compare (with non-mosaic) |Pair-wise Compare (with pure-random) |
> > > > |---|---|---|---|---|
> > > > | Pure-random Concatenation | 5.00 | 7.81 | 1.349 | 1.000 |
> > > > | Concatenation with Semantic Groups | 7.80 | 6.51 | 1.275 | 0.936 |
> > > >
> > > > We can draw insights from the above experiments:
> > > > 1. The **semantic concatenation can still outperform the non-mosaic baseline** by a large margin, indicating the effectiveness and potential of our Mosaic-IT augmentations and tasks.
> > > > 2. The semantic concatenation method has a slightly lower performance than the pure-random concatenation method, on pair-wise comparison and Alpaca Eval 2 scores. However, it achieves a much higher Alpaca Eval 2 (LC) score. This result suggests that **the response quality of the model trained with semantic concatenation is on par with pure-random but the response length is shorter and more condensed**.
> > > > 3. We found the semantic grouping leads to clusters with highly different average lengths of samples: The largest average length is 316.7 tokens while the smallest is 31.4 tokens. This discrepancy makes the lengths of Mosaic-IT concatenated samples more diverse, resulting in a better trade-off between quality and length of the responses.
> > > >
> > > > **Structure-based Grouping:**
> > > > Following your suggestion, we also tried grouping based on samples’ structures. Following common practice, we extracted the verb-noun pair of each instruction and tried to group the data samples by the verb-noun pair as it indicates the similarity of their targeted tasks. However, instructions in the modern instruction-tuning dataset are so diverse that most of the verb-noun pairs only appear a few times, thus it’s hard to group and conduct intra-cluster concatenation. We believe more structure-based grouping strategies can be investigated in the future.
> > > >
> > > > More experiments will be conducted and included in the future version of our paper. The main novel discovery we highlighted is that the **cost-free random concatenation already brings non-trivial improvement**. However, we agree that the random selection strategy may **not** be optimal, and more sophisticated strategies should be further investigated.

---

> > > > > ### Author Response · Authors · 2024-12-03
> > > > > **A kind summary of our comments for Reviewer cPom**
> > > > >
> > > > > Dear Reviewer cPom,
> > > > >
> > > > > Since the discussion period is about to end soon, we have prepared a concise summary of our responses to your last comment, focusing on the new updates from our side:
> > > > >
> > > > > **Q1: Why random concatenation is the optimal.**
> > > > >
> > > > > We did not claim that random concatenation is optimal. We chose it because it is entirely cost-free without requiring any extra prior knowledge or semantic grouping of data. The non-trivial improvement of such a straightforward application of the proposed augmentations is important in demonstrating the effectiveness of Mosaic-IT.
> > > > >
> > > > > **Q2: Structured or semantically grouped approaches.**
> > > > >
> > > > > We implemented the semantic grouping approach. Experimental results and conclusions are provided. Concatenating semantically similar samples to train LLMs can help generate condensed responses with comparable quality, when compared with pure-random concatenation, as reflected by the results on Alpaca Eval 2 (LC).
> > > > >
> > > > > We hope this summary can help you check whether and how we addressed the concerns you raised in your review and discussion. Based on the new updates, we would like to kindly inquiry you to consider raising the current rating to reflect our new results and improvement.
> > > > >
> > > > > Sincerely,
> > > > >
> > > > > Authors

---

> > > > > > ### Author Response · Authors · 2024-12-04
> > > > > > **A kind summary**
> > > > > >
> > > > > > Dear Reviewer cPom,
> > > > > >
> > > > > > As the discussion is about to end soon, we have prepared a concise summary of our responses to your last comment:
> > > > > >
> > > > > > **Q1: Why random concatenation is the optimal.**
> > > > > >
> > > > > > We did not claim that random concatenation is optimal. We chose it because it is entirely cost-free without requiring any extra prior knowledge or semantic grouping of data. The non-trivial improvement of such a straightforward application of the proposed augmentations is important in demonstrating the effectiveness of Mosaic-IT.
> > > > > >
> > > > > > **Q2: Structured or semantically grouped approaches.**
> > > > > >
> > > > > > We implemented the semantic grouping approach. Experimental results and conclusions are provided. Concatenating semantically similar samples to train LLMs can help generate condensed responses with comparable quality, when compared with pure-random concatenation, as reflected by the results on Alpaca Eval 2 (LC).
> > > > > >
> > > > > > We hope this summary can help you check whether and how we addressed the concerns you raised in your review and discussion. Based on the new updates, we would like to kindly inquiry you to consider raising the current rating to reflect our new results and improvement.
> > > > > >
> > > > > > Sincerely,
> > > > > >
> > > > > > Authors

---

### Official Review · Reviewer_mjBt · 2024-11-04

**Soundness:** 2
**Presentation:** 2
**Contribution:** 2
**Rating:** 5
**Confidence:** 5

**Summary:**

This paper studies instruction-tuning methods in LLMs by augmenting training data with three different templates, Format, Permute, and Maskout strategies. These techniques may reduce the over-fitting or memorization. The proposed method, Mosaic-IT, achieves consistent performance improvements over various benchmarks.

**Strengths:**

- [S1] The experimental results seems to be solid by demonstrating consistent improvement against a no-augmentation baseline.

**Weaknesses:**

- [W1] The techniques to prevent over-fitting and memorization by preparing various formatted templates and order randomization has been well studied and widely known approach; such as pioneering work of Instruction Tuning (Wei et al., 2021, Flan-T5 paper). From that time, the input/output pairs for instruction tuning are not always fixed and dynamically randomized. Considering these literatures, I think this paper is a kind of re-invention of those techniques, and the technical novelty and contribution seems to be limited.
- [W2] Figure 3 is unclear to me. Could you clarify what is a “mixture count”? While “Fix” strategy is adopted, the number of “mixture count” seems to be distributed among 1-10 (not fixed?). Why do you use Uniform as a default despite its not the best performance?
- [W3] In Figure 4, we can see that Mosaic-iT accelerates its training, but the performance at the convergence seems to be the same or even worse than the baselines, which is contradictory to your main results that improves the performance against baselines. Could you clarify the relationship between the convergence performance and the logic of performance improvement.
- [W4] In Table 4 (a) (ablation of Mosaic-IT), you tried Format, Permute, Maskout, and Permute/Maskedout. Why didn’t you try all the combinations?
Also, your best performance came from Maskout, but the adopted variant for Table 1 seems Permute/Maskedout. Why didn’t you use Maskout only?

**Reference**
- Wei et al., 2021. Finetuned Language Models Are Zero-Shot Learners. https://arxiv.org/abs/2109.01652

**Questions:**

See the weakness above.

---

> ### Author Response · Authors · 2024-11-24
> **Response to Reviewer #2(mjBt) (Part1)**
>
> **Weakness:**
>
> >Q1: The techniques to prevent over-fitting and memorization by preparing various formatted templates and order randomization has been well studied and widely known approach; such as pioneering work of Instruction Tuning (Wei et al., 2021, Flan-T5 paper).
>
> This paper is indeed an excellent pioneering work of Instruction Tuning, which greatly influences us. However, with all due respect, we **disagree** with your viewpoints on our paper and the Flan-T5 paper.
>
> 1. There is a **misunderstanding of the novelty and contribution** of our work. Our main contribution is **a cost-free compositional data augmentation method**, which concatenates existing instruction-tuning samples into complex ones. We did not utilize different templates for training. To our best knowledge, we are the first to propose this type of method for improving the instruction-following capability of LLMs, **please let us know if you find any reference applying similar ideas to LLMs.**
> 2. **The Flan-T5 work does not utilize or mention any idea of concatenating multiple instructions into one to make it more complex**, which is our main contribution and motivation.
> 3. **The instruction tuning setting of modern LLM in this paper is different from the one used in the Flan-T5 era**. Flan-T5 needs to cover various tasks and curate a lot of samples per task,  thus requiring different formats of templates to prevent overfitting. However, in the modern instruction tuning settings, every instruction is regarded as a different task and it does not require many samples per task, so the diverse templates are no longer required. Our work is based on the modern LLM instruction tuning setting.
>
> Please kindly let us know if you have any feedback.
>
> >Q2.1: Figure 3 is unclear to me. Could you clarify what is a “mixture count”?
>
> The mixture count is **the number of original samples/instructions to be composited in Mosaic-IT augmentations.** For example, in Figure 2, the mixture counts for the examples are all 3. We will further clarify it for more times in the paper.
>
> >Q2.2: While “Fix” strategy is adopted, the number of “mixture count” seems to be distributed among 1-10 (not fixed?).
>
> When “Fix” is applied, we fix the number of samples-to-be-composited to 10. However, some composited samples might exceed the maximum token limit for the instruction tuning. In this case, we decrease the number of samples to reduce the length below the limit. Otherwise, these augmentations need to be abandoned and some original samples are ruled out from the training. We mentioned this process in line 249, we will further modify the narratives to make it clearer.
>
> >Q2.3: Why do you use Uniform as a default despite its not the best performance?
>
> 1. Uniform distribution is not the best but it does not require search or tuning of the distribution and it already brings non-trivial improvement to the LLM. Trying more advanced distributions is optional but requires extra cost. We present the performance of other distributions to show the potential additional improvement that can be achieved by further tuning of distributions.
> 2. Uniform distribution by default keeps the method simple and the ablation studies clearer to understand, e.g., the ablation for the max number of instructions in Table 4 and the time reduction in Table 7.
>
> >Q3: In Figure 4, we can see that Mosaic-iT accelerates its training, but the performance at the convergence seems to be the same or even worse than the baselines, which is contradictory to your main results that improve the performance against baselines. Could you clarify the relationship between the convergence performance and the logic of performance improvement.
>
> Thank you for your comment, but we **disagree** on using **losses (especially training losses) to evaluate the performance of instruction tuning**.
>
> 1. The evaluation of instruction-following capability of LLMs is an open challenge in the community. Perplexity-based loss is not a reliable metric for evaluating the capability achieved by instruction tuning.
> 2. The data in the baseline method are entirely different from ours: our data is the concatenation of the baseline’s data, which is much more challenging for LLMs to learn. Thus their loss values are not comparable.

---

> ### Author Response · Authors · 2024-11-24
> **Response to Reviewer #2(mjBt) (Part2)**
>
> >Q4.1: In Table 4 (a) (ablation of Mosaic-IT), you tried Format, Permute, Maskout, and Permute/Maskedout. Why didn’t you try all the combinations?
>
> We evaluated the mix of Permute/Maskout strategies by applying them to different samples, as illustrated in line 381. We did not apply all strategies together to one sample in order to avoid potential ambiguities of the meta-instructions and misinterpretations of LLMs.
>
> For example, for a Mosaic-IT instruction “Ins1, Ins2, Ins3, Ins4, Ins5”, if we combine Permute and Maskout together, the meta-instruction can be “Respond to the instructions with the order of [3,2,1,5,4]. Ignore the 2nd and 3rd instructions.” Then there might be a misunderstanding on which instructions to ignore (the 2nd and 3rd of the original order or the permuted order?)
>
> More discussions will be included in our paper.
>
>
> >Q4.2 Also, your best performance came from Maskout, but the adopted variant for Table 1 seems Permute/Maskedout. Why didn’t you use Maskout only?
>
> Training LLMs to follow instructions in a predefined order is more challenging than ignoring some instructions. However, this skill might be more important than maskout as it is common to instruct LLMs to execute codes/instructions in a predefined order in practical applications.

---

> ### Author Response · Authors · 2024-11-25
> **A kind reminder**
>
> Dear Reviewer mjBt,
>
> As we are approaching the deadline of the discussion period, we would like to cordially inquire about the extent to which we have successfully addressed the concerns outlined in your review. Should there be any lingering points that require further attention, please rest assured that we are enthusiastic about the opportunity to provide comprehensive responses to any subsequent queries or comments you may have.
>
> Your constructive input remains invaluable to us, and we appreciate your dedication to enhancing the quality of our manuscript. Thank you for your time and consideration.
>
> Best,
>
> Authors

---

> ### Comment · Reviewer_mjBt · 2024-11-27
>
> Thank you authors for the detailed response and additional experiments. Also, I really sorry for the late reply due to my personal matter.
>
>
> I realized that my misunderstanding of the main focus of this paper. This paper aims to improve the capability of instruction-following under multiple rules, rather than general LLM's capability. I agree with the effectiveness of compositional combination of instructions to improve the instruction-following performance, and based on it I raised the score.
>
> My remaining concern is the evaluation. I don't think Alpaca Eval 2, MT-Bench, and Huggingface Open LLM Leaderboard are the suitable test bed to measure the capability of instruction-following, such as "Respond in reverse of the original order", "Ignore the longest one/several task(s) according to the word count.", "Enclose each reply with [START] and [END].", etc. While this method trains LLMs with those instructions, the capability of instruction-following with such instructions is not directly measured. I think the strong experimental performance in the relevant benchmarks does not appropriately support the claim on instruction-following capability.
>
> In addition, for the Figure 4, the author said,
>
> ```
> Thank you for your comment, but we disagree on using losses (especially training losses) to evaluate the performance of instruction tuning.
>
> The evaluation of instruction-following capability of LLMs is an open challenge in the community. Perplexity-based loss is not a reliable metric for evaluating the capability achieved by instruction tuning.
> The data in the baseline method are entirely different from ours: our data is the concatenation of the baseline’s data, which is much more challenging for LLMs to learn. Thus their loss values are not comparable.
> ```
>
> If this is true, why Figure 4 is presented in the paper? or it might better to remove it from the paper? The faster loss decrease does not related to the performance improvement. I think because we cannot compare the loss, the statement on the training efficiency does not make sense. The y-axis should be the performance on the instruction-following tasks strictly evaluated with whether the given instruction is satisfied or not.

---

> > ### Author Response · Authors · 2024-11-29
> > **Follow-up Response to Reviewer #2(mjBt)**
> >
> > We sincerely appreciate your time and effort in evaluating our manuscript and providing valuable feedback! In the following, we will respond to your latest concerns.
> >
> > >Q1: My remaining concern is the evaluation. I don't think Alpaca Eval 2, MT-Bench, and Huggingface Open LLM Leaderboard are the suitable test bed to measure the capability of instruction-following, such as "Respond in reverse of the original order", etc.
> >
> > Though our evaluations in the paper focus on the widely used instruction-following benchmarks, **we agree with you on the importance of verifying our trained models’ capability to follow the meta-instructions in Mosaic-IT**. To this end, we create a test set of compositional instructions from WizardLM test sets using Mosaic-IT. For simplicity, we name this new test setting as Mosaic task, which evaluates LLMs’ capability to follow multiple instructions with additional diverse constraints (meta-instructions).
> >
> > ----
> > **Here is an example**:
> >
> > Respond to each of the following instructions in reverse of the original order.
> >
> > [Ins1]
> >
> > [Ins2]
> >
> > [Ins3]
> >
> > ----
> >
> > We use the success rate (%) to evaluate the performance of models on the Mosaic task. A response is successful if it follows the meta-instruction and no instruction is ignored (unless the meta-instruction masks it). In the table below, we report the success rate (%) of LLMs following three meta-instruction strategies, i.e., Format / Permute / Maskout, on compositional augmentations of different numbers of instructions (i.e., 3, 5, 7 instructions). We report the success rates of GPT4o, two base models, and their Mosaic-IT finetuned versions.
> >
> > | Model                                  | 3 Instructions               | 5 Instructions               | 7 Instructions               |
> > |------------------|------------------|-------------------------------|-------------------------------|
> > | GPT4o                                  | 59.17 / 55.05 / 41.46         | 56.88 / 51.38 / 26.13         | 29.82 / 37.16 / 24.27         |
> > | | | | |
> > | Mistral + Alpaca-GPT4 (baseline)       | 20.18 / 3.67 / 3.25           | 10.09 / 2.75 / 5.41           | 7.34 / 0.92 / 0.97            |
> > | Mistral + Alpaca-GPT4 (mosaic)         | 98.32 / 66.51 / 69.11         | 95.87 / 60.55 / 67.57         | 97.25 / 64.68 / 66.02         |
> > | | | | |
> > | Llama3 + Magepie (baseline)            | 16.06 / 8.26 / 7.32           | 9.63 / 1.38 / 5.41            | 5.50 / 2.75 / 3.88            |
> > | Llama3 + Magepie (mosaic)              | 97.71 / 79.82 / 84.55         | 94.95 / 72.94 / 77.48         | 76.61 / 61.01 / 85.44         |
> >
> > The results expose the weaknesses of existing LLMs on Mosaic-IT tasks and show that training on Mosaic-IT augmentations can significantly improve performance. Specifically,
> >
> > 1. **Existing LLMs, even GPT4o, can not perfectly follow multiple instructions with diverse constraints**, not to mention other open-source models like Llama3  finetuned on datasets such as Magpie. These results further demonstrate the difficulty and complexity of Mosaic-IT tasks for existing LLMs, indicating the novelty of our method.
> > 2. **The compositional reasoning capability required by Mosaic-IT tasks cannot be covered by the capabilities of base LLMs and existing instruction-tuning datasets**. For example, the success rates of Mistral + Alpaca-GPT4 (baseline) and Llama3 + Magepie (baseline)  are similar, although Llama3 + Magepie has relatively better general instruction-following capabilities among them.
> > 3. **Our method can bridge the significant gap and enhance LLMs’ capability to follow multiple instructions with diverse constraints**. Moreover, our data augmentation is cost-free and does not take any effort from humans or models.
> >
> > Due to the limited time and space, we will include the full details of this experiment and evaluations with a more detailed analysis in our next version.
> >
> > >Q2: Why Figure 4 is presented in the paper? or it might better to remove it from the paper?
> >
> > Figure 4 is to discuss the potential memorization issue indicated by the shape of loss curves. It has been discussed in the community [1,2] that the losses for instruction tuning drop suddenly after each epoch (stair-like loss curves). This is probably caused by LLMs’ memorization of the training data, as the same data will be seen multiple times without any changes in training. This may hurt the generalization. In contrast, **our compositional augmentation creates diverse data that take different combinations of different original samples, so LLMs are always trained on different data which mitigates the memorization issue**. We do not intend to claim our method as the only one that can handle the memorization problem. Instead, we claim this is another merit of our method in addition to the improved performance and efficiency. We will clarify it better in our manuscript.
> >
> > [1] https://github.com/tatsu-lab/stanford_alpaca/issues/236
> >
> > [2] https://github.com/huggingface/transformers/issues/18730

---

> > > ### Author Response · Authors · 2024-12-01
> > >
> > > Dear Reviewer mjBt,
> > >
> > > We greatly appreciate your following-up discussion and glad to see that the original misunderstanding has been resolved. To address your further concerns on the evaluation, we provided new evaluation results above demonstrating the significant improvement achieved by our method on the meta-instruction following of multiple rules. Would you please check the results and let us know if you have any remaining concerns?
> > >
> > > Thanks!
> > >
> > > Authors

---

> > > > ### Author Response · Authors · 2024-12-03
> > > > **A kind summary of our comments for Reviewer mjBt**
> > > >
> > > > Dear Reviewer mjBt,
> > > >
> > > > Thank you for your thoughtful comments! As the discussion is about to end soon, we have prepared a concise summary of our responses to your last comment:
> > > >
> > > > **Q1: For the remaining evaluation concern.**
> > > >
> > > > We designed a new test set to be used for directly evaluating our models’ capability to follow multiple instructions with additional diverse constraints. Experimental results and conclusions are provided, showing our method's advantages.
> > > >
> > > > **Q2: For the presents of Figure 4.**
> > > >
> > > > We explained the reason why this figure is included.
> > > >
> > > > We hope this summary can help you check whether and how we addressed your concerns. Based on the new updates, we sincerely inquiry if you would like to consider increasing the current rating to reflect the latest improvement to the paper.
> > > >
> > > > Sincerely,
> > > >
> > > > Authors

---

> > > > > ### Author Response · Authors · 2024-12-04
> > > > > **A kind summary**
> > > > >
> > > > > Dear Reviewer mjBt,
> > > > >
> > > > > As the discussion is about to end soon, we have prepared a concise summary of our responses to your last comment:
> > > > >
> > > > > **Q1: For the remaining evaluation concern.**
> > > > >
> > > > > We designed a new test set to be used for directly evaluating our models’ capability to follow multiple instructions with additional diverse constraints. Experimental results and conclusions are provided, showing our method's advantages.
> > > > >
> > > > > **Q2: For the presents of Figure 4.**
> > > > >
> > > > > We explained the reason why this figure is included.
> > > > >
> > > > > We hope this summary can help you check whether and how we addressed your concerns. Based on the new updates, we sincerely inquiry if you would like to consider increasing the current rating to reflect the latest improvement to the paper.
> > > > >
> > > > > Sincerely,
> > > > >
> > > > > Authors

---

### Official Review · Reviewer_w54h · 2024-11-04

**Soundness:** 3
**Presentation:** 3
**Contribution:** 3
**Rating:** 6
**Confidence:** 4

**Summary:**

The paper argues that acquiring instruction-tuning data from a teacher model or humans is resource-intensive. In addition, it suggests that the complexity of single instruction can be limited for many instances which limits the instruction-following capabilities. To address this, the authors propose Mosaic-IT, a data augmentation strategy where the model is trained to follow multiple instructions via a meta instruction. Specifically, the paper considers multiple mosaic strategies including primary, maskout, permute, and format. Finally, the paper shows good improvements across models, instruction-tuning datasets and evaluation methods.

**Strengths:**

- The paper proposes an interesting way to stack multiple instructions to teach more complex instruction-following capabilities to them. It is encouraging that the authors consider many ways in which the instruction-response data can be stacked.
- The paper performs a diverse set of experiments across many base language models, instruction-tuning datasets, and evaluation methods.
- The paper performs several ablation studies to understand the usefulness of different experimental components.  The paper further analyzes the usefulness of the method using the smoothness of the learning dynamics.

**Weaknesses:**

- Motivation: how much of instruction tuning data acquisition is a bottleneck? There are several papers that show that a small number of instruction tuning data is enough to enable strong instruction-following capabilities. With the rise of powerful small language models (e.g., 4o-mini, Gemini-Flash, Haiku), getting a lot of instruction tuning data is not a bottleneck in terms of resources. In addition, I do not understand the connection between instruction tuning and Dense and Aligned Captions paper from the VL literature. The authors should rethink the motivation in the introduction. It is unclear whether this strategy scales with data i.e., having more Mosaic-IT data beneficial or not.
- The absolute performance on Alpaca2-LC seems too low. According to the original leaderboard [1], the AlpacaEval LC performance of Alpaca 7B (w/ LLama-1) is 5.9%, and Vicuna is 6.3%. However, the paper indicates that the baseline performance with much stronger base models (Mistral and LLaMA-3-8B) and datasets (Alpaca-GPT4, Wizard-70K, Vicuna, Magpie) is quite low. This makes me wonder if the models have been instruction tuned properly or not.
- Table 2 suggests that baseline methods have better 2-round MT-Bench scores than Mosaic-IT. Shouldn’t the second round MT-Bench scores improve with Mosaic-IT augmentation? Mosaic-IT shares similarity with multi-turn chats in the sense that both require answering multiple instructions in the given context.

[1] https://tatsu-lab.github.io/alpaca_eval/

**Questions:**

Mentioned in the weakness

---

> ### Author Response · Authors · 2024-11-24
> **Response to Reviewer #1(w54h) (Part1)**
>
> >Q1.1: Motivation: how much of instruction tuning data acquisition is a bottleneck? There are several papers that show that a small number of instruction tuning data is enough to enable strong instruction-following capabilities.
>
> This is still an open problem in the community. There are papers supporting that “a small number of instruction tuning data is enough for instruction-following capabilities”, including LIMA [1], Alpagasus [2], Cherry LLM [3], etc. However, there is also a very recent paper [4] mentioning data filtering is not effective for large-scale data. We will include this discussion in our later version.
>
> However, the discussion of this problem is out of the main scope of our paper, as we do not filter or discard any given data. **Our motivation and method are orthogonal to data filtering and data synthetic methods.** We aim to achieve cost-free data augmentations to further exploit existing instruction datasets without data filtering. The main difference and advantage of our method is that our data augmentation does not require any other models and is cost-free.
>
> >Q1.2 With the rise of powerful small language models (e.g., 4o-mini, Gemini-Flash, Haiku), getting a lot of instruction tuning data is not a bottleneck in terms of resources.
>
> Powerful small language models can reduce the generation cost of instruction tuning data more easily, however:
>
> 1. Our method avoids the entire cost of data synthesis by any language models or neural networks. In contrast, small language models (SLMs) still produce concerning carbon footprints.  Their lower cost per sample may come with a price of more trials. For example, to achieve superalignment (SLM-generated data improves LLMs), strategies like best-of-n are often needed but significantly increase the cost.
> 2. Our method aims to **exploit the potential of existing data** so the quality of augmented data is 100% guaranteed. In contrast, the newly generated data by SLMs still require careful and complex quality checks in practice to reduce the risk of degrading the original LLM (i.e., negative finetuning).
> 3. **Mosaic-IT is orthogonal and complementary to existing data filtering and data synthesis approaches.** It can be applied together with other methods to further improve LLMs.
> 4. As a model-free method, Mosaic-IT can avoid **the potential violation of licenses** limiting the usage of existing LLMs.
>
> >Q1.3: I do not understand the connection between instruction tuning and Dense and Aligned Captions paper from the VL literature. The authors should rethink the motivation in the introduction.
>
> Thank you for your insightful suggestion! We used the idea of “dense alignment” from the “Dense and Aligned Captions” paper to motivate the importance and challenges of generating multiple responses aligning with the corresponding instructions in the input according to the meta-instruction in Mosaic-IT. In the cited paper, learning with dense captions helps the model to achieve the capability of aligning each part of the image with corresponding descriptive captions. This is similar to the dense alignment between multiple instructions and responses in the compositional data by Mosaic-IT. We will modify the motivation to make it clearer.
>
>
> >Q1.4:  It is unclear whether this strategy scales with data i.e., having more Mosaic-IT data beneficial or not.
>
> Our experiments include datasets with various data sizes including 50k, 70k, 300k, and 1M (results shown in Q2). Training with Mosaic-IT data achieves consistently better performances across these datasets.

---

> ### Author Response · Authors · 2024-11-24
> **Response to Reviewer #1(w54h) (Part2)**
>
> >Q2: The absolute performance on Alpaca2-LC seems too low.
>
> Thank you for reading our paper in such detail!
>
> For finetuning on Vicuna 1M data, we randomly selected 300K data for the training due to the computation budgets, considering the diverse data quality of Vicuna 1M, (which contains conversations of other languages and some dummy conversations like “Hello!” “Hello!”), thus we think the main cause of the low performance is the random selection. Thus, we further finetuned Llama-3-8B with all the 1M data to see how the performance goes, the results are shown below, which is much better:
>
> | Model + Dataset | Alpaca Eval 2 (LC) |  Alpaca Eval 2  |
> |---|---|---|
> |Llama-3-8B + Vicuna 1M | 8.6% | 8.4%|
> |Llama-3-8B + Vicuna 1M + Mosaic | 9.7% | 9.1% |
>
> For finetuning on Magepie data, we also used the filtered 300K data. All the training settings are kept the same as reported in the paper, except for the maximum sequence length, which we set to 4096 compared with their 8192 for the computation budget. We further finetuned Llama-3-8B with the 8192 length  to see how the performance goes, the results are shown below, which is slightly better:
>
> | Model + Dataset | Alpaca Eval 2 (LC) |  Alpaca Eval 2  |
> |---|---|---|
> |Llama-3-8B + Magepie 300K | 18.4% | 20.8%|
> |Llama-3-8B + Magepie 300K + Mosaic | 20.5% | 22.6% |
>
> Despite the absolute performance variances between models finetuned by us and the reported performances, the comparison between the baseline and our method is totally under the same setting to ensure a fair comparison, which we believe is solid for verifying the effectiveness of our methods.
>
> >Q3: Table 2 suggests that baseline methods have better 2-round MT-Bench scores than Mosaic-IT. Shouldn’t the second round MT-Bench scores improve with Mosaic-IT augmentation?
>
> 1. Mosaic-IT does not necessarily improve the second-round dialogue of LLMs if the instruction data are single-round conversations. Our method composites several instructions into one but it is still under the setting of single round conversations.
> 2. The second-round performances are affected by both the instruction data and the LLMs to be trained. As shown in Table 3, when more advanced data and LLMs are used, the second-round performances can be further improved.
>
> More discussion will be included in the later version.
>
>
> [1] LIMA: Less Is More for Alignment. (NeurIPS’23)
> [2] AlpaGasus: Training a Better Alpaca with Fewer Data. (ICLR’24)
> [3] From quantity to quality: Boosting llm performance with self-guided data selection for instruction tuning. (NAACL’24)
> [4] Rethinking Data Selection at Scale: Random Selection is Almost All You Need.

---

> ### Author Response · Authors · 2024-11-25
> **A kind reminder**
>
> Dear Reviewer w54h,
>
> As we are approaching the deadline of the discussion period, we would like to cordially inquire about the extent to which we have successfully addressed the concerns outlined in your review. Should there be any lingering points that require further attention, please rest assured that we are enthusiastic about the opportunity to provide comprehensive responses to any subsequent queries or comments you may have.
>
> Your constructive input remains invaluable to us, and we appreciate your dedication to enhancing the quality of our manuscript. Thank you for your time and consideration.
>
> Best,
>
> Authors

---

> > ### Comment · Reviewer_w54h · 2024-11-26
> > **Response to authors**
> >
> > Hi,
> >
> > I thank the authors for the clarifications, and it addresses some of my concerns. Please include them in the future versions of the paper. I have increased my rating.

---

> > > ### Author Response · Authors · 2024-11-27
> > >
> > > Thank you for your reply! We will surely include all the discussions in the later version.

---

### Author Response · Authors · 2024-11-24
**General Response**

We sincerely appreciate the time and effort the reviewers had taken to evaluate our manuscript and provide valuable feedback. In the following, we will respond to the major concerns.

>Q1: Why our method works?

1.
**Mosaic-IT trains LLMs to follow meta-instructions for compositional reasoning.**

Previous methods train LLMs to produce a response for a single instruction or query. Instead, our method produces compositional data augmentations to **train LLMs to generate multiple responses for multiple instructions in diverse forms** (e.g., order, mask, format) specified by different meta-instructions. It also enforces LLMs to partition the input context correctly and manage the interference and dependencies among multiple instructions. These are critical to developing and improving the compositional reasoning capabilities of LLMs, which have not been covered by mainstream instruction-tuning frameworks.

2.
**Mosaic-IT creates more challenging and complex instructions to further improve LLMs’ instruction-following capabilities.**

Mosaic-IT’s composition of multiple instructions and the diverse meta-instructions create more challenging and complex instruction-tuning data for LLMs. Moreover, since we do not rely on data synthesis using LLMs but solely apply some rules to existing data, the correctness and quality of the augmented data are guaranteed. As shown in Section 5.1, even powerful LLMs like GPT4 can not follow concatenated instructions. **It has been widely accepted that such challenging and complex instructions improve LLMs’ instruction-following capability [1-8].  Mosaic-IT follows this intuition by making the instruction more challenging and complex** in order to improve LLMs. Different from previous methods relying on humans or stronger teacher LLMs to create the challenging samples, Mosaic-IT does not require any humans/models to create the augmentations.

To quantitatively evaluate the difficulty and complexity of instruction-tuning data, [2] proposes a ChatGPT-based method (Number of InsTag), while [5] proposes a perplexity-based Instrutcion-Following Difficulty (IFD) score. We compute these two metrics on the Alpaca and WizardLM70k datasets to verify the effectiveness of our method in improving the difficulty/complexity:

**Number of InsTag [2]:**
The number of InsTag is used to measure the complexity of the instructions. A larger value of the Number of InsTag indicates the intentions of the instruction are complex and benefit the LLM instruction tuning process. For the experiments below, we prompt GPT4o with the exact prompt provided in [2] to generate the Instags.

Average InsTag (Alpaca): 2.62

Average InsTag (Alpaca-Mosaic): 9.75

Average InsTag (WizardLM): 4.20

Average InsTag (WizardLM-Mosaic): 10.93

Mosaic-IT largely increases the average number of InsTag, indicating a large increase in instruction intention complexity, further leading to better performance.

**IFD score [5]:**
IFD score is a perplexity-based metric used to evaluate the instruction-following difficulty of a given instruction-response pair. A higher IFD score indicates that it is hard for the current model to build a connection between the instruction and the corresponding response, so it can be used to select training data beneficial for LLM instruction tuning. For the experiments below, we utilized the IFD score computed on GPT2.

Average IFD (Alpaca): 0.60

Average IFD (Alpaca-Mosaic): 0.76

Average IFD (WizardLM): 0.67

Average IFD (WizardLM-Mosaic): 0.79

Mosaic-IT increases IFD scores, indicating an increase in the instruction-following difficulty, which leads to an improvement in performance.

[1] LIMA: Less Is More for Alignment. (NeurIPS’23)

[2] #InsTag: Instruction Tagging for Analyzing Supervised Fine-tuning of Large Language Models. (ICLR’24)

[3] WizardLM: Empowering Large Pre-Trained Language Models to Follow Complex Instructions. (ICLR’24)

[4]What Makes Good Data for Alignment? A Comprehensive Study of Automatic Data Selection in Instruction Tuning. (ICLR’24)

[5] From quantity to quality: Boosting llm performance with self-guided data selection for instruction tuning. (NAACL’24)

[6] Superfiltering: Weak-to-strong data filtering for fast instruction-tuning. (ACL’24)

[7] Selective reflection-tuning: Student-selected data recycling for llm instruction-tuning. (ACL’24)

[8] Instruction Fusion: Advancing Prompt Evolution through Hybridization. (ACL’24)

>Q2: Novelty and contribution

To the best of our knowledge, Mosaic-IT is the **first cost-free compositional data augmentation for instruction tuning of LLMs**. It reduces the training cost and simultaneously improves the performance. In contrast, most existing data-enhancement methods for instruction tuning rely on human supervision or additional LLMs to generate new data.

---

### Meta-Review · Area_Chair_2VLd · 2024-12-22

**Metareview:**

The paper introduces a data augmentation method for instruction-tuning, the most common post pretraining stage that makes LLMs most useful. The authors propose Mosaic-IT, which can create diverse augmentations from samples of existing instruction tuning datasets. The method works by randomly concatenating multiple instructions into one and requiring the LLM to follow meta-instructions.
The method overcomes the issue of requiring a strong(er) teacher model to rewrite instruction datasets and shows reductions in training time. The idea is also clear and simple and motivated from computer vision's mosaic augmentation strategy for object detection.
However, the paper in its current form has severe weaknesses: the mosaic strategies are somewhat arbitrary, the training time reduction in instruction tuning is typically not essential, as this step is very short (~13h) on 4 GPUs and similarly whether the current issue of IT data is indeed the quantity (which if it were the case, would motivate Mosaic IT). While the AC disagrees with w54c that the ubiquity of industrial (private) IT datasets counteracts the value of the proposed paper, the point does stand. Moreover, specifying the augmentation format to single-turn conversation does seem to lead to a decrease in performances in 2-round MT Bench.

**Additional Comments On Reviewer Discussion:**

The message to the AC has been considered. While the authors provided rebuttals, the reviewers did engage in some discussion (mjBt, cPom). Yet the points raised remain: the authors further agree that 2-round MT-Bench results are affected, for some LLMs, without much discussion. Similarly the points of mosaicing strategies being arbitrary and lack of analysis of why random concatenation works best remain. Combined with the point of the lack of a clear movitation regarding the need for Mosaic IT in current IT setups, this paper is just below the high bar of acceptance for ICLR and the AC recommends rejection.

---

### Decision · Program_Chairs · 2025-01-22

Reject